# *ESR1* mutant breast cancers show elevated basal cytokeratins and immune activation

Zheqi Li[1,2,3], Olivia McGinn[2,3], Yang Wu[2,3,4], Amir Bahreini[2,3,5], Nolan M. Priedigkeit[1,2,3], Kai Ding[2,3], Sayali Onkar[2,3,6,7,8], Caleb Lampenfeld[6,7,8], Carol A. Sartorius[9], Lori Miller[2,3], Margaret Rosenzweig[10], Ofir Cohen[11,12], Nikhil Wagle[11,12], Jennifer K. Richer[9], William J. Muller[13], Laki Buluwela[14], Simak Ali[14], Tullia C. Bruno[6,7,8], Dario A. A. Vignali[6,7,8], Yusi Fang[15], Li Zhu[15], George C. Tseng[15], Jason Gertz[16], Jennifer M. Atkinson[1,2,3], Adrian V. Lee[1,2,3,5] & Steffi Oesterreich[1,2,3,5✉]

Estrogen receptor alpha (ER/*ESR1*) is frequently mutated in endocrine resistant ER-positive (ER+) breast cancer and linked to ligand-independent growth and metastasis. Despite the distinct clinical features of *ESR1* mutations, their role in intrinsic subtype switching remains largely unknown. Here we find that *ESR1* mutant cells and clinical samples show a significant enrichment of basal subtype markers, and six basal cytokeratins (BCKs) are the most enriched genes. Induction of BCKs is independent of ER binding and instead associated with chromatin reprogramming centered around a progesterone receptor-orchestrated insulated neighborhood. BCK-high ER+ primary breast tumors exhibit a number of enriched immune pathways, shared with *ESR1* mutant tumors. S100A8 and S100A9 are among the most induced immune mediators and involve in tumor-stroma paracrine crosstalk inferred by single-cell RNA-seq from metastatic tumors. Collectively, these observations demonstrate that *ESR1* mutant tumors gain basal features associated with increased immune activation, encouraging additional studies of immune therapeutic vulnerabilities.

[1] Department of Pharmacology and Chemical Biology, University of Pittsburgh, Pittsburgh, PA, USA. [2] Womens Cancer Research Center, UPMC Hillman Cancer Center, Pittsburgh, PA, USA. [3] Magee-Womens Research Institute, Pittsburgh, PA, USA. [4] School of Medicine, Tsinghua University, Beijing, China. [5] Department of Human Genetics, University of Pittsburgh, Pittsburgh, PA, USA. [6] Department of Immunology, University of Pittsburgh, Pittsburgh, PA, USA. [7] Cancer Immunology and Immunotherapy Program, UPMC Hillman Cancer Center, Pittsburgh, PA, USA. [8] Tumor Microenvironment Center, UPMC Hillman Cancer Center, Pittsburgh, PA, USA. [9] Department of Pathology, University of Colorado Anschutz Medical Campus, Aurora, CO, USA. [10] School of Nursing, University of Pittsburgh, Pittsburgh, PA, USA. [11] Department of Medical Oncology and Center for Cancer Precision Medicine, Dana-Farber Cancer Institute, Harvard Medical School, Boston, MA, USA. [12] Department of Medicine, Brigham and Women's Hospital, Boston, MA, USA. [13] Goodman Cancer Centre and Departments of Biochemistry and Medicine, McGill University, Montreal, QC, Canada. [14] Department of Surgery and Cancer, Imperial College London, Hammersmith Hospital Campus, London, UK. [15] Department of Biostatistics, University of Pittsburgh, Pittsburgh, PA, USA. [16] Department of Oncological Sciences, Huntsman Cancer Institute, University of Utah, Salt Lake City, UT, USA. ✉email: oesterreichs@upmc.edu

Breast cancer is characterized by a high degree of heterogeneity, originally identified through the use of immunohistochemistry and gene expression profiling[1,2]. Broadly, molecular subtypes can be grouped into luminal (luminal A and luminal B), HER2-enriched and basal-like tumors, primarily driven by expression of ER, PR, HER2 and Ki67[3]. Tumors with different molecular subtypes show distinguishing clinical features and therapeutic responses[4,5], including metastatic spread and immune profiles[6,7].

The basal-like subtype, which represents 15–25% of all cases and overlaps with triple negative breast cancers (TNBC), is characterized by a unique gene expression profile similar to that of myoepithelial normal mammary cells[8]. Basal-like breast cancers are more aggressive and patients suffer from shorter metastases-free survival compared to those with luminal subtypes[8,9]. Mechanisms underlying increased invasive properties of basal-like tumors include deregulation of the CCL5/CCR5 axis[10], amplified EGFR[11] kinase signaling and activation of TGF-β signaling[12]. Despite multiple signaling aberrations providing challenges for efficient therapeutic strategies, recent studies have unveiled unique vulnerabilities of basal-like breast cancers, such as higher levels of PD-L1 expression along with constitutive IFNγ signaling activation[13], in line with higher immune-infiltration scores[6]. While the FDA has granted an accelerated approval for atezolizumab, a monoclonal antibody drug targeting PD-L1, plus chemotherapy for the treatment of TNBC[14], the potential application of immune therapies for patients with luminal breast cancer remains largely unknown.

Among the four intrinsic subtypes, basal and luminal subtypes show opposite histochemical features and notable differences in prognosis[15,16], however there is increasing evidence that these subtypes are on a continuum of "luminal-ness" and "basal-ness" features. Models of breast cancer lineage evolution describe that basal and luminal progenitor cells are derived from the same bipotential progenitors[17], indicating the potential of lineage reprogramming during cancer progression. Such subtype switching during tumor evolution has been described and is critical for implementation of precision therapeutics[18–20]. A recent study by Bi et al. reported loss of luminal and gain of basal markers in endocrine resistant breast tumors[21]. Mechanisms underlying the intrinsic subtype plasticity are largely unknown, with some exceptions. JARID1B[22] and ARID1A[23] have been described as essential luminal lineage driver genes and their mutations result in luminal-to-basal subtypes switches. In addition, enhancer reprogramming at GATA3 and AP1 binding sites has been highlighted as a pivotal epigenetic mechanism allowing lineage plasticity[21].

ER is well characterized as a luminal lineage marker[24]. Hotspot mutations in its ligand-binding domain occur in 30–40% of endocrine resistant breast tumors, promoting ligand-independent ER activation and metastasis[25–27]. Several recent studies showed that ESR1 mutant tumors are not only associated with endocrine resistance, but also gain unexpected resistance towards CDK4/6 inhibitors[28], mTOR inhibitors[29] and radiation therapy[30] in a mutation subtype and context dependent manner, suggesting potentially more complex re-wiring of ER mutant tumors.

Here, we set out to examine whether ESR1 mutations alter the "luminal-ness" and "basal-ness" balance in breast cancer cell line models and clinical specimens. We discover that ER mutant tumors gain basal-like features, characterized by elevated expression of basal cytokeratins as a result of epigenetic reprogramming. Immune context analyses in clinical specimens reveal potential therapeutic vulnerabilities accompanying the increased basal-ness in ESR1 mutant breast cancer, a finding of potential clinical relevance.

## Results

**Basal gene signatures are enriched in ESR1 mutant breast cancer.** To examine whether ESR1 mutations alter "luminal-ness" and "basal-ness" we utilized five independent luminal and basal gene signatures (Fig. 1a and Supplementary Data 1). Gene sets from Charafe-Jauffret et al.[31] and Huper et al.[32] were obtained from MSigDB (Supplementary Fig. 1a, b), and in addition we generated three other gene sets from i) intrinsic subtype genes[33] differentially expressed between luminal ($n = 33$) and basal ($n = 39$) breast cancer cell lines (Supplementary Data 2)[34–36] and ii) genes differentially expressed between luminal and basal primary tumors in TCGA[37] and METABRIC[38] (Supplementary Fig. 1c–e). Although the overlap among the different gene sets was limited (Fig. 1b), likely reflecting differences in methodology and sources, some well described lineage marker genes (e.g., ESR1 and FOXA1 as luminal markers, and KRT6A and KRT16 as basal markers) were observed in 4 out of 5 gene sets.

As expected, all five basal gene sets were significantly enriched in basal versus luminal breast cancer cell lines and tumors (Supplementary Fig. 2a, b), and vice versa for luminal gene sets except for the Huper luminal markers, likely due to its derivation from normal mammary tissue (Supplementary Fig. 2c, d). We found concordantly increased enrichment of basal gene sets in Y537S and D538G MCF7 ESR1 genome-edited mutant cells, whereas no differences were observed in estrogen treated ESR1 wildtype cells (Fig. 1c). In contrast, we did not observe a consistent change in the luminal gene sets (Fig. 1d). This was further corroborated by PAM50-based analysis, where we found MCF7 ESR1 mutant cells were predominantly called as basal subtype with above 70% probability (Supplementary Fig. 3a) and exhibited gene expressional similarities to basal breast cancer cell lines (Supplementary Fig. 3b). The enrichment of the basal gene sets in the ESR1 mutant cells was also seen in an independent CRISPR-engineered MCF7 ESR1 mutant cell model recently reported by Arnesen et al.[39] (Supplementary Fig. 3c) and in our T47D ESR1 mutant cells[27] (Supplementary Fig. 3d). Of note, no consistent and strong alterations of luminal and basal gene sets enrichment levels were detected in ESR1 WT endocrine resistant ER+ breast cancer cell models[21,40–47] (eight tamoxifen resistant, two fulvestrant resistant, and seven long-term estradiol deprivation (LTED) models), suggesting that the "basal-ness" shift is a unique feature acquired as a result of ESR1 mutations (Supplementary Fig. 3e).

We next sought to extend our findings to clinical specimens using RNA-seq data composed of 51 intra-patient matched ER+ primary-metastatic tumor pairs (7 ESR1 mutant and 44 ESR1 WT pairs) (Supplementary Table 1). Similar to observations in cell lines, ESR1 mutant metastatic breast cancers showed a significant enrichment of basal gene signatures compared to tumors with WT ESR1 (Fig. 1e). We did not observe a concurrent decrease of luminal markers (Fig. 1f). Taken together, these findings suggested a unexpected gain of "basal-ness" in ESR1 mutant tumors.

**Basal cytokeratins are elevated in ESR1 mutant breast cancer cells and tumors.** We next interrogated the union of the five basal gene sets ($N = 742$) to identify which basal marker genes were consistently induced in ESR1 mutant breast cancer cells. Integrating RNA-seq results from MCF7 cell models[27] and clinical samples identified a group of basal cytokeratins (BCKs) (KRT5, KRT6A, KRT6B, KRT14, KRT16, and KRT17) as the top consistently increased basal markers (Fig. 2a and Supplementary Fig. 4a and Supplementary Table 2). Elevated basal cytokeratins mRNA levels were further confirmed in independent qRT-PCR experiment in ESR1 mutant MCF7 cells (Fig. 2b). Analyzing

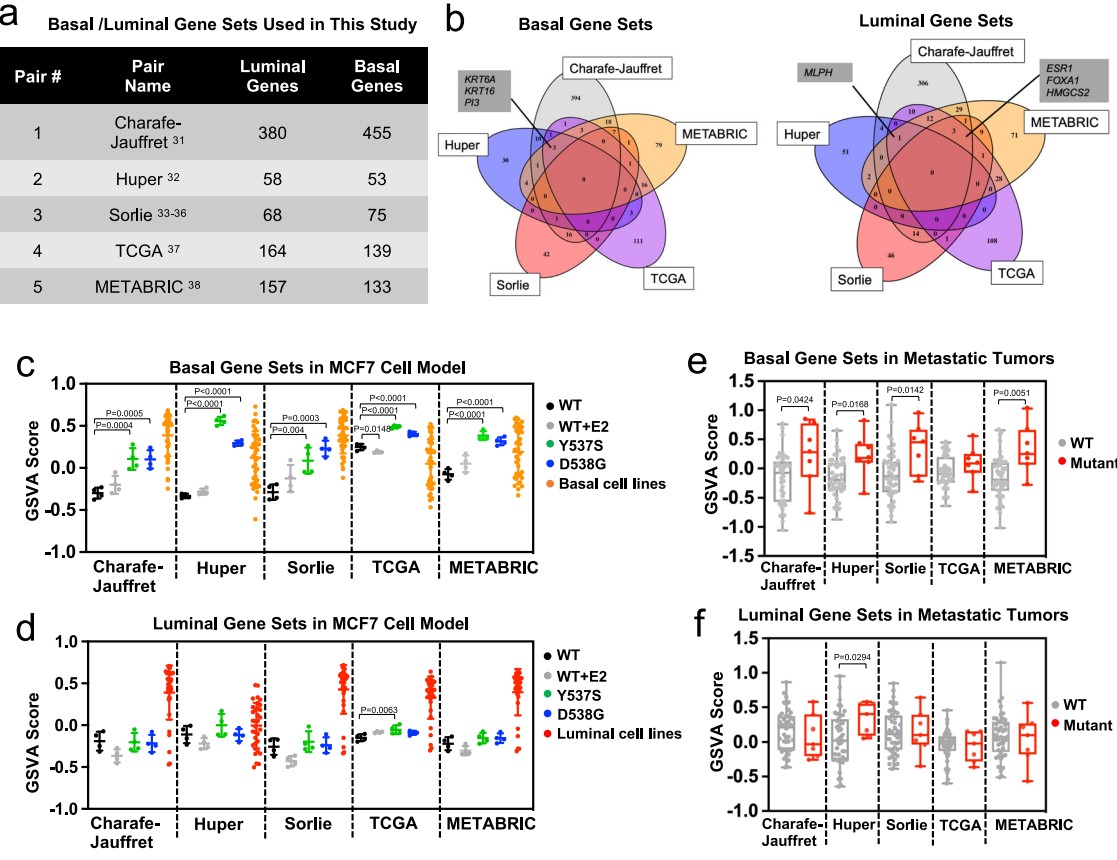

**Fig. 1 Basal breast cancer gene sets are enriched in *ESR1* mutant breast cancers. a** Five pairs of luminal/basal gene sets applied in this study with gene numbers specified in each set. **b** Venn diagram representing the overlap of genes from the basal (left) and luminal (right) gene sets. Genes overlapping in at least four gene sets are indicated. **c**, **d** Dot plots showing GSVA score of the five pairs of basal (**c**) and luminal (**d**) gene sets enrichment in MCF7 genome-edited cell models. Four biologically independent replicates were used from the original RNA-seq data set (GSE89888) for one time computation. Scores from luminal ($n = 33$) and basal ($n = 39$) breast cancer cell lines were used as positive controls. Data are presented as mean ± SD. Dunnett's test (two-sided) was used to compare with WT-vehicle set within each gene set. **e**, **f** Box plots representing basal (**e**) and luminal (**f**) gene set enrichments in intra-patient matched paired primary-metastatic samples. Delta GSVA score for each sample was calculated by subtracting the scores of primary tumors from the matched metastatic tumors. Box plots span the upper quartile (upper limit), median (center) and lower quartile (lower limit). Whiskers extend a maximum of 1.5× IQR. Mann–Whitney $U$-test (two-sided) was performed to compare the Delta GSVA scores between WT ($N = 44$) or *ESR1* mutation-harboring ($N = 7$) paired tumors. Source data are provided as a Source Data file for **c**–**f**.

fold-change expression of all basal markers in a number of MCF7 *ESR1* mutant cell models previously described[25,27,39] revealed *KRT5,16* and *17* as the top increased basal genes (Supplementary Fig. 4b–d). In the T47D *ESR1* mutant cells, *KRT16* was significantly increased (Supplementary Fig. 4e), but the observed enrichment of basal marker genes (Supplementary Fig. 3d) was also driven by other non-canonical basal genes such as *WLS and HTRA1* (Supplementary Table 3), suggesting some context-dependent mechanisms for the increased basalness.

We also queried *KRT* expression in overexpression models. In MCF7 cells with stable overexpression of HA-tagged WT and mutant ER (Y537S and D538G) (Supplementary Fig. 5a and 5b), we again observed significant overexpression of *KRT5, KRT6A, KRT6B, KRT16*, and *KRT17* (Supplementary Fig. 5c).

Given higher BCK mRNA expression in *ESR1* mutant cells, we examined their expression at the protein level. We confirmed higher CK5 and CK16 protein levels in early passage (P6-8) *ESR1* mutant cells, but curiously expression was not detectable in later passages (P30-32) (Supplementary Fig. 6a). This finding was consistent with prior reports on slower growth of CK5+ sub-populations[48], reflecting selection forces eliminating BCK-positive subclones from luminal cell populations. To determine whether BCK expression was limited to minor sub-populations in *ESR1* mutant cells, we performed IF staining for CK5, CK16, and

CK17 in early passage cells (below P12) (Fig. 2c). No BCK positive clones were identified in MCF7-WT cells, while 0.5–1% of Y537S and D538G *ESR1* mutant cells exhibited strong diffuse cytoplasmic CK5/16/17 expression. In addition, 3–5% of *ESR1* mutant cells displayed strong BCK signals localized as foci adjacent to the nucleus (Supplementary Fig. 6b), and this was again not observed in the WT cells. Furthermore, co-staining of CK5 + CK16 and CK16 + CK17 showed that the BCK proteins were predominantly (in 75–90% imaged cells) upregulated in the same sub-population of cells (Supplementary Fig. 6c, d). In contrast, luminal cytokeratin CK8 was homogenously expressed with stronger expression at the edges of each cell cluster (Supplementary Fig. 6e), suggesting that the marked heterogeneity was a unique feature for BCK expression in the luminal cell background. Importantly, the heterogenous expression of CK5 and CK17 was confirmed in an ER+ liver metastatic lesion harboring an Y537S mutation (Fig. 2e, f and Supplementary Fig. 4f).

**BCK induction is independent of mutant ER DNA binding but requires low ER expression.** Mutant ER can function in a ligand-independent manner[26,27], and we thus tested whether induction of BCKs resulted from ligand-independent ER activity. We

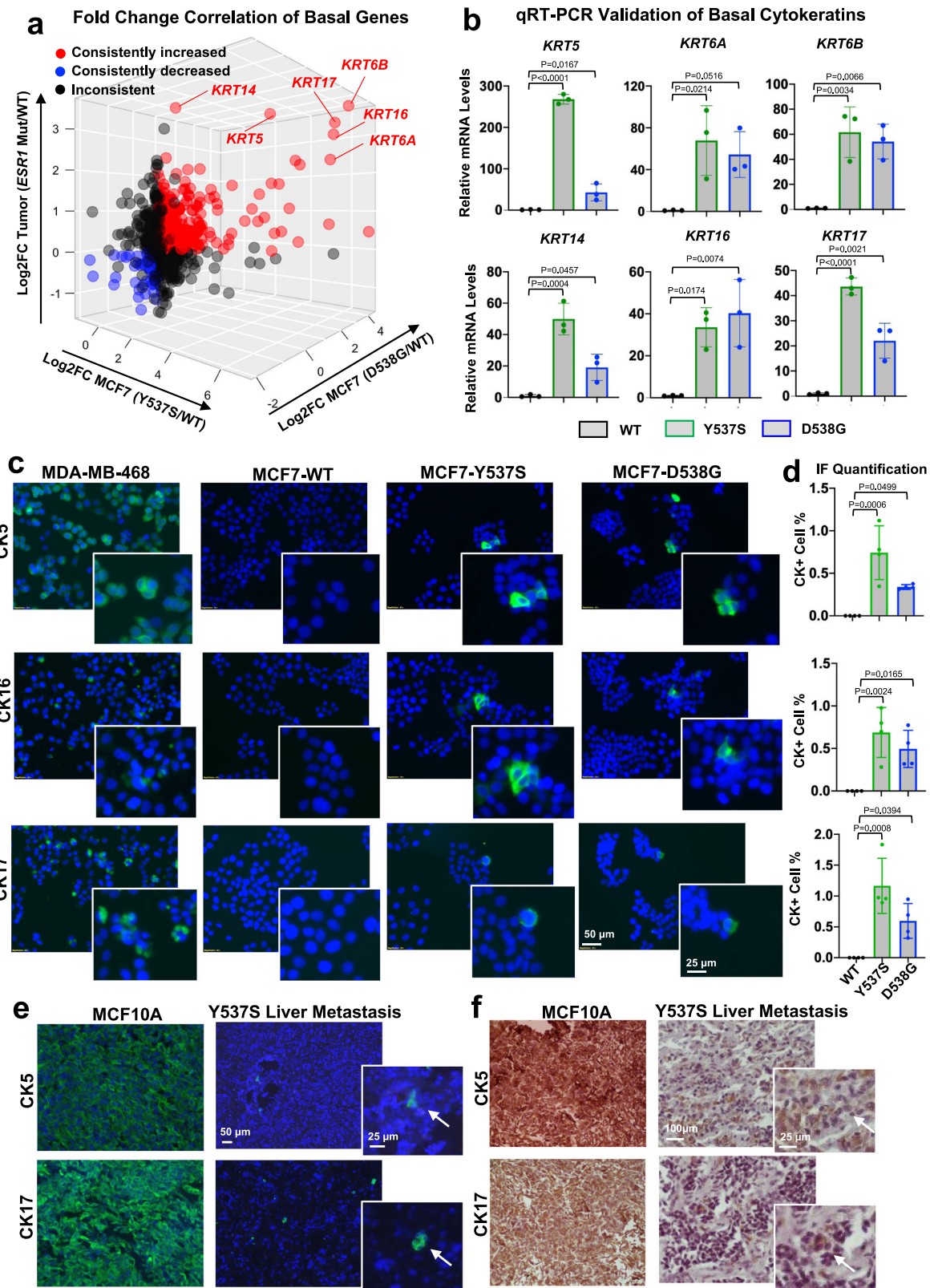

interrogated eight publicly available RNA-seq and microarray data sets with estradiol (E2) treatment in six different ER+ breast cancer cell lines[26,27,49–52]. In contrast to strong E2 induction of classical ER target genes such as *GREB1, TFF1,* and *PGR*, expression of basal and luminal cytokeratins genes was not regulated by E2 with the exception of *KRT7* (Fig. 3a). We then examined whether BCK expression was regulated via de novo genomic

binding of mutant ER at BCK genes. We performed ChIP-seq in MCF7 WT and *ESR1* mutant cells in the absence and presence of E2. As expected, in the absence of E2 we detected very few ER binding sites in WT MCF7 cells ($n = 125$), whereas E2 stimulation triggered substantial ER binding events ($n = 12,472$) (Supplementary Table 4). Consistent with previous studies[25,26], Y537S and D538G ER show strong ligand-independent binding, with 657

**Fig. 2 Overexpression of basal cytokeratins (BCK) in *ESR1* mutant breast cancer cells and tumors. a** Correlation between basal gene fold changes (FC) in MCF7-Y537S/D538G cells (normalized to WT vehicle) and intra-patient paired mutant tumors (normalized to WT tumors) (*N* = 742). Consistently increased or decreased genes in the two MCF7 mutant cells and tumors compared to their WT counterparts were highlighted in red or blue respectively, and six basal cytokeratin genes are indicated. Inconsistently changed genes among the three comparisons are labeled in black. **b** *KRT5/6A/6B/14/16/17* mRNA levels in MCF7 WT and *ESR1* mutant cells. Relative mRNA fold change normalized to WT cells and *RPLP0* levels measured as the internal control. Each bar represents mean ± SD with three biological replicates. Representative results from three independent experiments are shown. Dunnett's test (two-sided) was used to compare BCKs expression levels between WT and mutant cells. **c** Representative images of immunofluorescence staining on CK5, CK16, and CK17 in MCF7 WT and *ESR1* mutant cells. Regions with CK positive cells were highlighted in the magnified images. MDA-MB-468 was included as positive control. Images were taken under ×20 magnification. **d** Quantification of percentages of CK positive cells in MCF7 WT and *ESR1* mutant cells. Each bar represents mean ± SD from four different regions. Data shown are from one representative experiment of three independent experiments. Dunnett's test (two-sided) was used to compare BCKs positive cell prevalence between WT and mutant cells. **e, f** Immunofluorescent (**e**) and immunohistochemistry (f) staining of CK5 and CK17 on sections from MCF10A (positive control) and a Y537S *ESR1* mutant liver metastasis tissue. Images were taken under ×10 (IF) or ×20 (IHC) magnification. Subclones with CK5 or CK17 expression were further magnified and highlighted with white arrow. This experiment was done once on clinical specimens. Source data are provided as a Source Data file for **a**, **b**, **d**.

binding sites in Y537S and 1016 in D538G mutant cells (Supplementary Fig. 7a). The *GREB1* gene locus is shown as a representative example (Fig. 3b, left panel). Furthermore, intersection analysis with other reported ChIP-seq data sets of *ESR1* mutant cells revealed considerable intersection ratios (Supplementary Fig. 7b)[25,26,39], despite some inter-model variations. Co-occupancy analyses between WT-E2 and mutant-vehicle sets demonstrated that one third of all Y537S (36%) and D538G (31%) ER binding sites were not detected in the WT + E2 data suggesting gain-of-function binding sites (Supplementary Fig. 7c); however, none of them mapped to the BCKs genes with increased expression in *ESR1* mutant cells (−/+ 50 kb of transcriptional start sites) (Fig. 3b, middle and right panel). This was further corroborated in four additional ER ChIP-seq data sets in MCF7 *ESR1* mutant cell models[25,26,39] (Supplementary Fig. 7d).

We then expanded our analyses and examined potential estrogen-regulation of all basal marker genes, again using the union of the five basal gene sets (*N* = 742). Comparison of E2 and *ESR1* mutation-conferred fold changes of these genes in MCF7 cells revealed that the top upregulated basal markers in *ESR1* mutant cells were not E2-induced (Supplementary Fig. 7e, f). In addition, only 23 basal genes (3%) harbor mutant ER binding sites at −/+ 50 kb of TSS (Supplementary Fig. 7g), and 20 of those were not differentially expressed between WT and mutant cells (Supplementary Fig. 7h). Taken together, these analyses suggest that the shift to "basal-ness" in *ESR1* mutant cells was not mediated via ligand-independent binding of mutant ER to BCK gene loci.

To further understand interplay between *ESR1* and *KRT* gene expression, we determined expression of basal and luminal *KRT* genes in ER+ primary breast tumors. As shown in Fig. 3c, the six BCKs were significantly negatively correlated with *ESR1* expression, whereas the luminal *KRT* were mostly positively correlated with *ESR1* (Fig. 3c). Luminal *KRT7* was again the exception, being negatively correlated with *ESR1* expression, in line with it being repressed by ER (Fig. 3a). The inverse correlation between BCK and *ESR1* expression was also reflected in results from ER knockdown experiments, in which loss of *ESR1* significantly increased expression of BCKs in MCF7 WT and mutant cells (Fig. 3d). Similar results were obtained in five additional ER+ breast cancer cell lines, where we observed a general increase of BCK expression after *ESR1* knockdown (Supplementary Fig. 8). In addition, co-staining of ER and CK5/CK16/17 in MCF7 *ESR1* mutant cells showed significantly lower ER expression in BCK+ cells than in the surrounding BCK- cells (Fig. 3e). Collectively, these data demonstrate that ER serves as a negative regulator of BCKs expression independent of ligand and mutational status, and suggest that low ER expression is likely necessary but not sufficient to facilitate BCKs overexpression in a subpopulation of

*ESR1* mutant cells. These data also support a role for mutant ER in regulating BCK expression via epigenetic regulation, a mechanism that we have recently shown to be used by mutant ER[39].

**PR regulation of BCK expression through binding at a CTCF-driven chromatin loop at the *KRT14/16/17* loci in *ESR1* mutant cells.** To investigate potential epigenetic regulation of *KRT5/6A/6B* and *KRT14/16/17*, we first compared their regional epigenetic landscapes on chromosome 12 and 17, respectively, in luminal and basal breast cancer cell lines and tumors (Supplementary Fig. 9). Integrative analysis of ATAC-seq and ChIP-seq profiles of H3K4me2, H3K4me3, H3K9ac, and H3K27ac suggested that these two regions are epigenetically silent in MCF7 (Supplementary Fig. 9a), consistent with low expression. In basal breast cancer cell lines and tumors, there is an enrichment of H3K27 acetylation (Supplementary Fig. 9b) and number of ATAC-seq peaks (Supplementary Fig. 9c) at BCK loci, consistent with increased mRNA expression (Supplementary Fig. 9d–f). This is also observed in *ESR1* mutant cell models (Supplementary Fig. 9g).

We recently reported CCCTC-binding factor (CTCF) motif as one of the top enriched motifs in unique *ESR1* mutant-regulated accessible genomic regions[39]. To determine whether CTCF has a role in the epigenetic regulation of BCK, we developed a CTCF gene signature by identification of the top 100 differentially expressed genes before and after CTCF knockdown in MCF7[53] (Supplementary Data 1). The positively correlated CTCF signature (i.e., using genes that were repressed after CTCF knockdown) was significantly enriched in both MCF7 *ESR1* mutant cells (Fig. 4a) and metastatic tumors (Fig. 4b) compared to their WT counterparts, whereas E2 stimulation had no effect (Fig. 4a). CTCF is a multimodal epigenetic regulator in breast cancer[54], in part through generating boundaries of insulated neighborhoods and guiding of DNA self-interaction[55]. Mapping the genomic occupancy of CTCF and three other cohesion complex members (RAD21, STAG1 and SMC1A) in MCF7 cells[56–58] (Fig. 4c) identified five putative insulated neighborhood boundaries at the *KRT14/16/17* (Fig. 4d) loci and three at the *KRT5/6A/6B* (Supplementary Fig. 10a) loci. Integration of an additional MCF7 CTCF ChIA-PET dataset[59] showed that a strong chromatin loop is predicted to span the *KRT14/16/17* genes, further supported by the pattern of convergent CTCF motif orientations at the predicted insulated neighborhood boundaries (Fig. 4c) and visualization of a Hi-C data set in MCF7 cell line[42] (Supplementary Fig. 10b). Since the *KRT5/6A/6B* locus did not harbor strong chromatin loops (>3 linkages), we focused our further analysis on the *KRT14/16/17* locus.

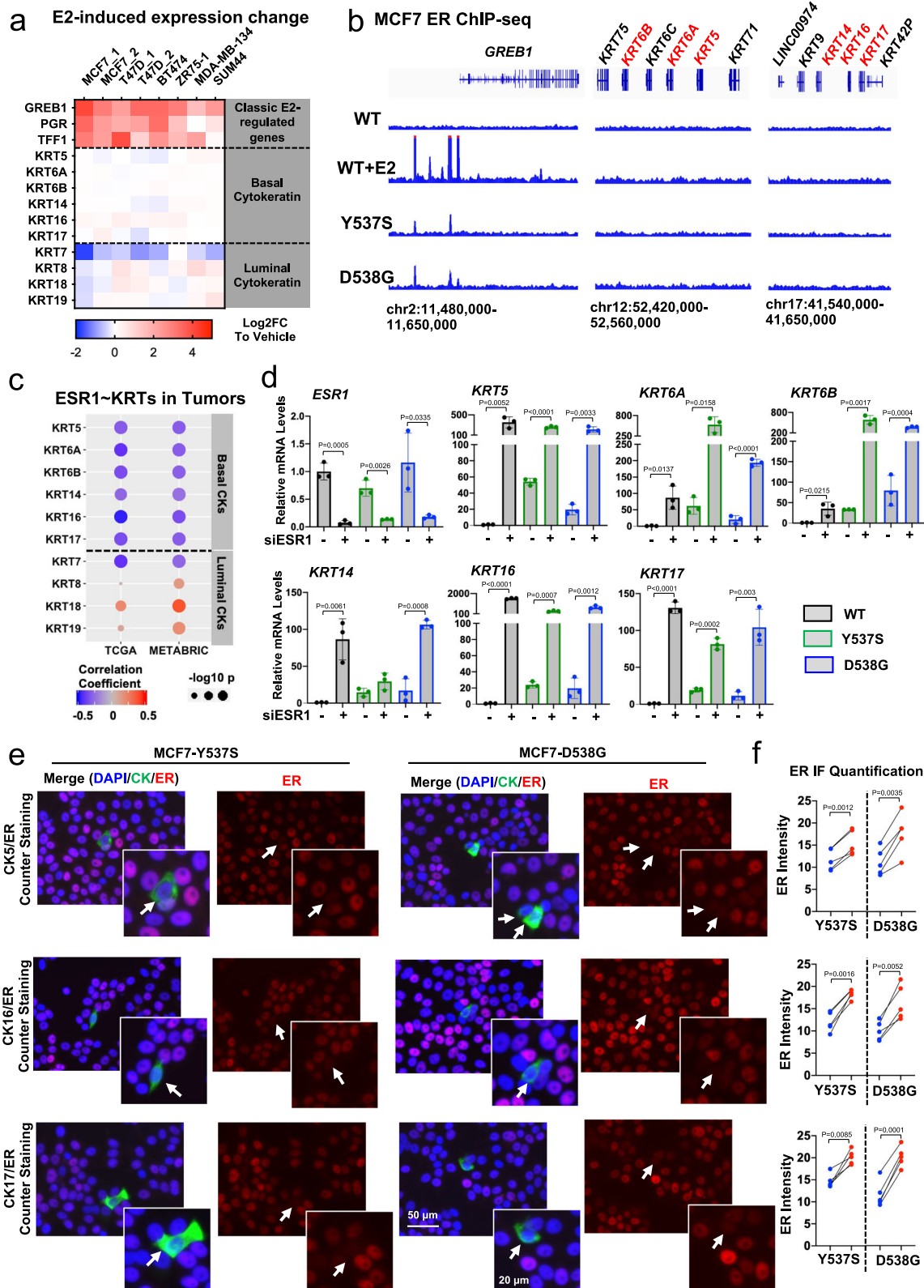

**a** E2-induced expression change

**b** MCF7 ER ChIP-seq

**c** ESR1~KRTs in Tumors

**d**

**e**

MCF7-Y537S   MCF7-D538G

**f** ER IF Quantification

ChIP revealed strong enrichment of CTCF binding at the base of the chromatin loops of the *KRT14/16/17* locus in *ESR1* mutant cells, however there was a lack of E2 regulation (Fig. 4e). Decreasing CTCF levels led to increased expression of *KRT14, KRT16*, and *KRT17* mRNA levels in *ESR1* mutant cells (Supplementary Fig. 10c), potentially reflecting a role for CTCF as "classical" insulator, suppressing high expression of these BCKs

through the identified super enhancer at the *KRT14, KRT16* locus (Fig. 4f). Given identification of progesterone receptor (PR) binding sites within this super enhancer, PR's previously identified role in regulating *KRT5* expression in luminal breast cancer cells[48,60], and finally its overexpression in multiple *ESR1* mutant cell models[25–27,61] (Supplementary Fig. 10d, e), we tested whether PR regulates *KRT14/16/17* expression.

**Fig. 3 Basal cytokeratins induction is independent of mutant ER genomic binding but requires low ER expression. a** Heatmap representing fold change mRNA expression (E2/veh) of six basal cytokeratins and four luminal cytokeratins in ER+ breast cancer lines from six publicly available data sets (GSE89888, GSE94493, GSE108304, GSE3834, GSE38132, and GSE50693). *GREB1*, *PGR*, and *TFF1* are canonical E2-regulated genes included as positive controls. **b** Genomic track showing ER binding intensities at *KRT5/6A/6B* and *KRT14/16/17* loci from ER ChIP-seq data sets of MCF7 *ESR1* mutant cells. *GREB1* locus serve as a positive control. **c** Graphic view of Pearson correlation between expression of *ESR1* and each basal or luminal cytokeratin in ER+ breast tumors in TCGA (*n* = 808) and METABRIC (*n* = 1505) cohorts. Color scale and size of dots represent correlation coefficient and significance, respectively. **d** qRT-PCR measurement of *ESR1*, *KRT5/6A/6B/14/16/17* mRNA levels in MCF7 WT and *ESR1* mutant cells with *ESR1* siRNA knockdown for 7 days. mRNA fold changes were normalized to WT cells; RPLP0 levels were measured as internal control. Each bar represents mean ± SD with three biological replicates. Data shown are representative from three independent experiments. Student's *t*-test (two-sided) was used to compare the gene expression between scramble and knockdown groups. **e** Representative images of ER, CK5, CK16, and CK17 staining in MCF7-Y537S and D538G cells. BCKs positive cells are highlighted with white arrows. Images were taken under ×20 magnification. **f** Dot plots quantifying the ER intensities in BCKs positive (blue) and the corresponding proximal negative (red) cells from each region. Individual data points from five different regions per group from one experiment, representative of three independent experiments are shown. Paired *t*-test (two-sided) was applied to compare ER intensities between BCKs positive and negative cells. Source data are provided as a Source Data file for **a**, **c**, **d**, **f**.

PR ChIP-seq revealed a ligand-inducible PR binding sites in MCF7 cells approximately 32 kb upstream of the *KRT14/16/17* loop region[62] (Fig. 4f). This PR binding site overlapped with a curated super-enhancer in MCF7 cells[63], which was additionally supported by strong active histone modifications (Supplementary Fig. 9). Knockdown of PR partially rescued the increased expression of *KRT14*, *16*, and *17* in both *ESR1* mutants (Fig. 4g, Supplementary Fig. 10f). We also observed a similar rescue effect for *KRT5* (Supplementary Fig. 10f), consistent with previous studies[60]. Identification of double positive (CK17+ and PR+) cells in a Y537S *ESR1* mutant patient-derived xenograft tumor (HCI-013EI)[64] provides further support for regulation of BCK by PR (Supplementary Fig. 10g). Furthermore, both PR agonist (P4) and antagonist (RU486) treatment increased *KRT5*, *16*, and *17* expression in Y537S *ESR1* mutant cells, while only RU486 triggered *KRT5* and *KRT16* expression in D538G mutant (Fig. 4h and Supplementary Fig. 10h). The marked induction effect of RU486, a PR antagonist, is likely due to its previously reported partial agonism via recruitment of coactivators[65]. The RU486-induced CK5 and CK16 increase was further examined by IF, where CK5 (Supplementary Fig. 10i) and CK16 (Fig. 4i, j) positive cells increased from 1 to 5%. Of note, CK17 positive cells were not increased by RU486 treatment (Supplementary Fig. 10j), suggesting translational efficiency differences between different BCK subtypes. Together, these data demonstrated that elevated PR expression in *ESR1* mutant cells was essential for BCKs induction, and this was possibly due to an orchestration with a super enhancer which is accessible to regulate *KRT14/16/17* genes via the CTCF-driven chromatin loop.

Since glucocorticoid receptor (GR, *NR3C1*) shares similar response motif with PR, we tested whether GR could also activate BCKs expression in *ESR1* mutant cells. Unlike the substantial overexpression of PR, GR expression was moderately repressed in *ESR1* mutant MCF7 cells (Supplementary Fig. 11a). Knockdown of GR increased expression of BCKs (except *KRT17*) in both *ESR1* WT and mutant cells (Supplementary Fig. 11b), and GR binding was identified at the super enhancer region at *KRT14/16/17* loci (Supplementary Fig. 11c). These data suggest that although GR can bind to regulatory regions in keratin genes, it is unlikely to play a causative role in BCK induction observed in *ESR1* mutant cells.

**Enhanced immune activation, associated with S100A8-S100A9 secretion and signaling in *ESR1* mutant tumors.** Finally, we investigated whether the increased expression of basal genes in *ESR1* mutant tumors confers basal-like features and potentially novel therapeutic vulnerabilities. To identify basal cytokeratin-associated pathways enriched in ER mutant tumors, we at first identified ER+ tumors with the top and bottom quantile of BCK

gene enrichment and then computed hallmark pathways differentially enriched between these two groups (Supplementary Fig. 12a). Intersection of these BCKs-associated pathways with those enriched in *ESR1* mutant metastases uncovered seven shared molecular functions, the top five of which are all related to immune responses (Fig. 5a and Supplementary Figs. 12b, S12c and Supplementary Table 5). An orthogonal approach—bioinformatic evaluation using ESTIMATE[66]—confirmed the unique enhancement of immune activation in BCK-high vs. BCK-low ER+ tumors which is not seen in ER-/HER2+ or TNBC subtypes (Supplementary Fig. 12d), albeit overall it is still lower than in basal tumors (Fig. 5b). In addition, BCK-high tumors displayed higher lymphocyte and leukocyte fractions according to a recent biospecimens report[67] (Fig. 5c), and higher *PDCD1* mRNA levels (Supplementary Fig. 12e). Intriguingly, patients with BCK-high ER+ tumors experience improved outcomes in univariable and multivariable analysis (Fig. 5d and Supplementary Fig. 12f), and although entirely speculative at this point in time, one could hypothesize that this might be due to increased anti-tumor immune activation.

Similar to BCK-high ER+ tumors, *ESR1* mutant metastatic tumors exhibited higher immune scores compared to those with *ESR1* WT (Fig. 5e). Immune cell subtype deconvolution[68,69] revealed significantly higher CD8+ T, NK, and dendritic cells, along with macrophages in *ESR1* mutant tumors. The higher CD8+ T cell scores were also observed in BCK-high primary tumors in TCGA and METABRIC (Supplementary Fig. 12g). In addition, immune checkpoint expression analysis revealed higher expression of *VISTA* in *ESR1* mutant tumors (Supplementary Fig. 12h), which has been characterized as a key suppressor of T cell-associated immune response in human cancer[70] and can be pharmacologically targeted[71]. Basal breast cancers harbor high immune infiltrations at least in part due to higher tumor mutation burden (TMB)[72], however, we did not detect higher TMB in BCK-high vs. low ER+ tumors (Supplementary Fig. 12i).

To understand which factors might contribute to immune activation in *ESR1* mutant and BCK-high ER+ tumors, we compared gene expression of major immune genes derived from ESTIMATE[73] (*n* = 141) between *ESR1* mutant and WT tumors, and BCK-high vs. BCK-low ER+ tumors. This analysis identified S100A8 and S100A9 as the two top consistently increased immune-related genes (Fig. 6a), and this overexpression was also seen in MCF7 *ESR1* mutant cell models (Supplementary Fig. 12j). S100A8 and S100A9 are pro-inflammatory cytokines that form heterodimers and play crucial roles in shaping immune landscapes[45,46]. As expected, S100A8/A9 expression correlated positively with immune scores in ER+ tumors (Fig. 6b), however, S100A8/A9 expression did not associate with improved survival, suggesting a more complex role in this context (Supplementary

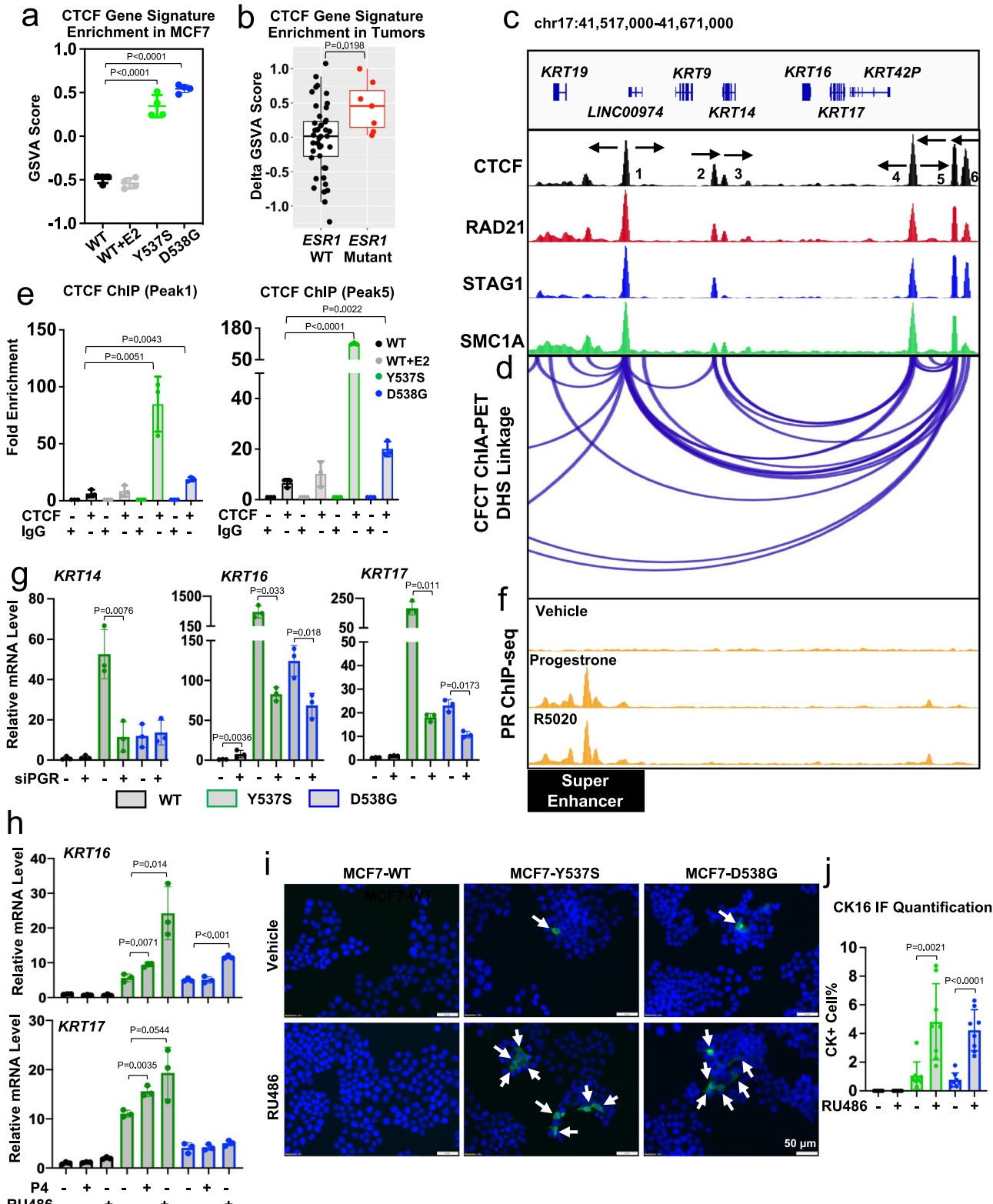

Fig. 12k). BCKs levels failed to differentiate immune scores in ER+ tumors among the subset of tumors exhibit high S100A8-A9 (Fig. 6b). S100A8/A9 are secreted proteins and function as heterodimers. To confirm S100A8/A9 protein overexpression, we measured S100A8/A9 heterodimer levels in plasma samples from patients with *ESR1* WT ($n = 7$) and mutant ($n = 11$) tumors (Supplementary Table 6) (Fig. 6c). This analysis revealed significantly higher circulatory S100A8/A9 heterodimers

concentrations in plasma from patients with *ESR1* mutations (Fig. 6d).

S100A8-A9 heterodimer mainly stimulates downstream cascades through two receptors: toll-like receptor 4 (TLR4) and receptor for advanced glycation end products (RAGE), and both of them are widely reported to impact cancer immunity. A further gene set variation analysis in WCRC/DFCI primary-matched paired metastatic samples revealed consistent enrichment of both

**Fig. 4 PR regulation of BCK expression through binding at a CTCF-driven chromatin loop. a, b** Enrichment levels of CTCF gene signature in MCF7 *ESR1* mutant cells (*n* = 4) (**a**) and *ESR1* WT (*n* = 44) and mutant (*n* = 7) metastases (**b**). Box plots span the upper quartile (upper limit), median (center) and lower quartile (lower limit). Whiskers extend a maximum of 1.5× IQR. Dunnett's test (two-sided) (**a**) and Mann–Whitney *U*-test (two-sided) was used. **c, d** Genomic track illustrating the CTCF and cohesion complex (**c**) binding and CTCF-driven chromatin loops (**d**) at *KRT14/16/17* proximal genomic region in MCF7 cells. CTCF motif orientations of each peak is labeled with arrows. *Y*-axis represents signal intensity of each track. **e** CTCF binding events at binding sites 1 and 5 in **c**. Each bar represents mean ± SD of fold enrichment normalized to IgG from three independent experiments. Pair-wise *t*-test (two-sided) was performed. **f** Genomic track view of PR ChIP-seq in MCF7 cells under R5020 and progesterone treatments. *Y*-axis represents signal intensity under the same scale. Super enhancer range is highlighted. **g** *KRT14*, 16, and 17 mRNA levels in MCF7 *ESR1* WT and mutant cells with *PGR* siRNA knockdown for 7 days. Each bar represents mean ± SD of fold changes normalized to WT cells with three biological replicates as a representative from three independent experiments. Student's *t*-test (two-sided) was used. **h** *KRT16* and 17 mRNA levels in MCF7 *ESR1* WT and mutant cells treated with 0.1% EtOH, 100 nM P4 or 1 μM RU486 for 3 days. Each bar represents mean ± SD of fold changes normalized to WT cells with three biological replicates as a representative from three independent experiments. Dunnett's (two-sided) test was used. **i** Representative images of immunofluorescence staining of CK5 and CK16 in MCF7 WT and *ESR1* mutant cells after 3 day treatment with 1% EtOH or 1 μM RU486. **j** Quantification of CK positive cells in **i**. Each bar represents mean ± SD from eight different regions combining from two independent experiments. Student's *t*-test (two-sided) was used. Source data are provided as a Source Data file for **a**, **b**, **e**, **g**, **h**, **j**.

---

pathways in *ESR1* mutant tumors (Fig. 6e and Supplementary Data 1), suggesting both TLR4 and RAGE signaling are hyperactive in *ESR1* mutant tumors. However, in vitro stimulation of human-derived macrophages with S100A8/S100A9 purified proteins failed to induce cytokine production (Supplementary Fig. 12l), possibly due to required interaction of the S100A8/S100A9 heterodimer with additional factors from the tumor microenvironment.

To further elucidate the specific cell–cell communication by S100A8/S100A9 signaling in the tumor ecosystem, we analyzed RAGE and TLR4 signaling via measuring ligand and receptor expression in different cell types using single-cell RNA-seq data from two breast cancer metastases. Highest expression of *S100A8/S100A9* was seen in epithelial cells, followed by fibroblast and macrophages (Fig. 6f, g). This was confirmed by immunofluorescent staining in the Y537S liver metastatic tissue, where we found S100A8/A9+ cells overlapped with approximately 60% and 40% EpCAM+ and CD45+ cells, respectively (Fig. 6h, i), validating that epithelial cancer cells are a prevalent source for S100A8/A9 production in the ecosystem. In contrast, *TLR4* and *AGER* showed low expression in the epithelial cells, but instead were widely expressed in the stroma, especially in fibroblasts and macrophages. In general, *AGER* displayed lower expression levels in all cell types compared to *TLR4* (Fig. 6f, g).

Taken together, these data support the concept that the increase in basal-ness of *ESR1* mutant tumors is associated with immune activation, in part facilitated by the paracrine S100A8/A9-TLR4 signaling.

## Discussion

Recurrence of ER+ breast cancer causes over 24,000 deaths each year in the US alone. Given that *ESR1* mutation occur in as many as 20–30% of metastatic recurrences, it is imperative to identify therapeutic vulnerabilities through dissecting mechanisms of action. In this study we have uncovered a previously unrecognized plasticity of *ESR1* mutant cells, reflected by enrichment of basal subtype genes in *ESR1* mutant tumors and in particular a gain of BCK expression, resulting from epigenetic reprogramming of a mutant ER-specific PR-linked chromatin loop. This molecular evolution, i.e., an increase of basal-like feature in the *ESR1* mutant tumors was associated with immune activation including enhanced S100A8/A9-TLR4 signaling (Fig. 7).

Increased plasticity of tumors has previously been shown to be associated with tumor initiation and progression[21,47,74–76]. PAM50 intrinsic subtype switching has been described to occur in as many as 40% of breast cancer metastases[20]. Here we show that *ESR1* mutant cells gain basal-ness, and a similar observation was recently reported by Gu et al.[77] showing a luminal to basal switch

in MCF7 *ESR1* Y537S CRISPR cells compared to parental cells. However, luminal to basal subtype switching is rare in breast cancer[20] and we have previously reported on clinically relevant gene expression changes in brain metastases (increased in *HER2* gene expression) without clear subtype switching[18]. These results are in line with the increasing appreciation of the molecular subtypes being on a continuum rather than representing discrete stages. Of note, we did not observe a similar gain of basal-ness in a series of *ESR1* wildtype endocrine resistant in vitro models, with the exception being a study revealing a "luminal-to-basal" switch in an estradiol-deprived T47D xenograft derived cell line, indicating a potential role for the microenvironment in mediating a similar switch in ER wildtype tumors[78].

We propose that the observed *ESR1* mutant-cancer cell state interconversions are of potential clinical relevance due to increased stromal immune activation associated with the induction of BCK. Using in silico gene expression, pathway analyses, and pathology information, we observed increased activation of a number of immune-related pathways including S100A8/S100A9-TLR4 signaling and increased lymphocytic infiltration. S100A8/S100A9 heterodimers exhibit pro-inflammatory properties in different contexts in breast cancer[79,80], are associated with poor prognosis in multiple cancer types[36] including breast cancer[81], and blockade of their activity improves survival[82]. We observed increased S100A8/S100A9 levels in blood from patients with *ESR1* mutant tumors but given complexity of tumor-cell intrinsic and extrinsic roles of the inflammatory mediators and their receptors (also supported by our single cell sequencing analysis) additional work is needed to understand if and how they contribute to tumor progression in patients with ER mutant tumors. Notably, we failed to detect effects of S100A8/S100A9 using an in vitro system, suggesting the need for more complex model systems including in vivo models. This will also allow the analysis of MDSC which have been described to play an important role in S100A8/A9 function[80,83]. This is also supported by our recent studies showing an enrichment of immune-suppressive macrophages in ER mutant tumors, along with increased expression of interferon regulated genes[84]. Additional multi-center studies are necessary to comprehensively characterize immune infiltration in *ESR1* mutant tumors, including the analysis of spatial heterogeneity, as a recent study demonstrates that spatial clustering of immune cells (Immune Spatial Score) is linked to poor recurrence-free survival in ER+ breast cancers[85]. Immune activation by S100A8/A9 may reshape the architecture of cancer-immune ecosystem. Nevertheless, our data suggest the enhanced immune activation in *ESR1* mutant breast cancers as a therapeutic vulnerability. There is data showing enhanced immune filtrations were associated with worse outcome of ER+ breast

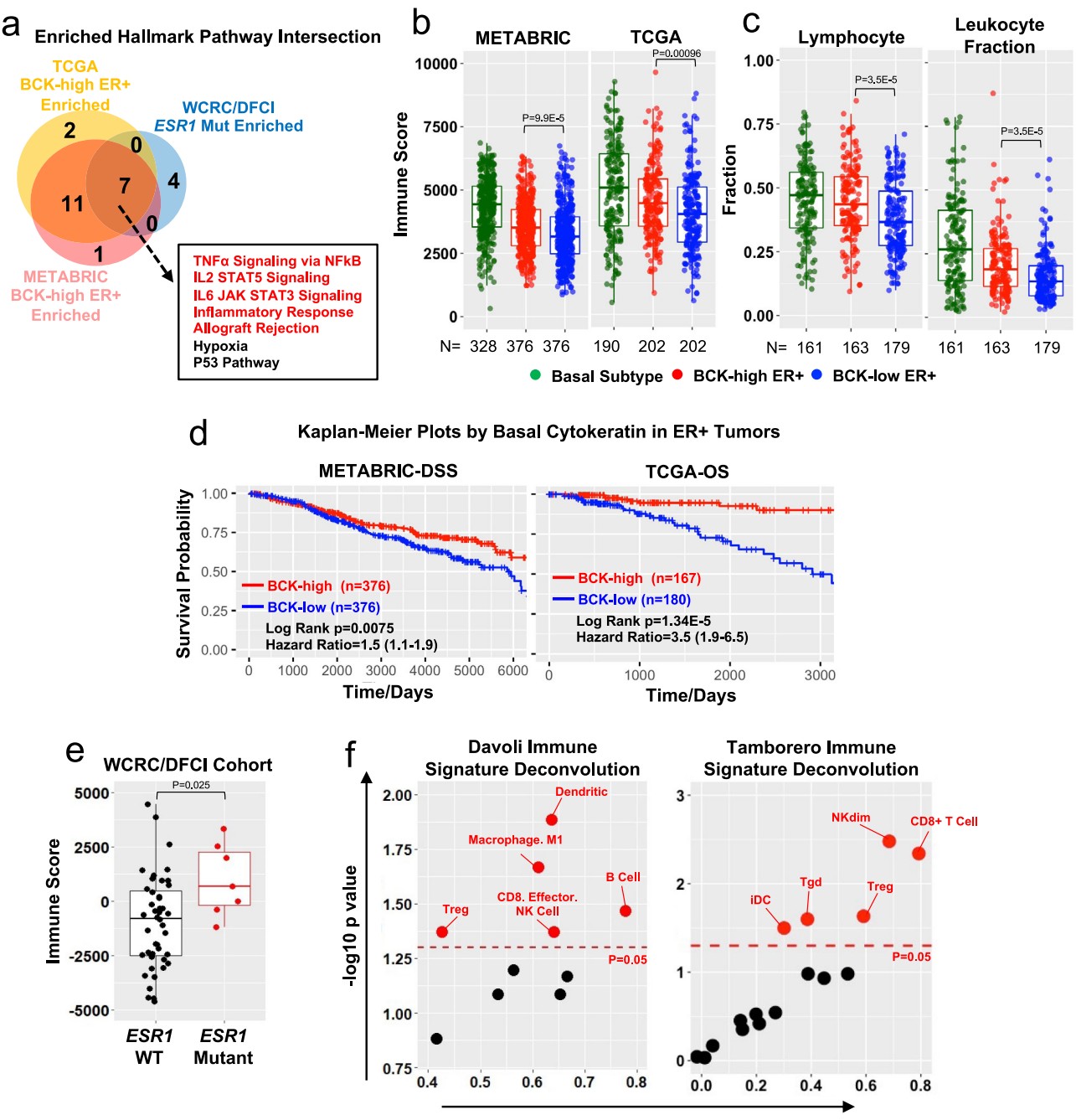

**Fig. 5 Gain of basal cytokeratin expression is associated with enhanced immune activation in *ESR1* mutant tumors. a** Venn diagrams showing the intersection of significantly enriched hallmark pathways in three sets of comparisons: BCK-high vs low in 1) TCGA ER+ tumors ($n = 202$ in each group), 2) METABRIC ER+ tumors ($n = 376$ in each group) and 3) *ESR1* mutant ($n = 7$) vs. WT ($n = 44$) metastatic tumors. BCKs high and low were defined by the upper and bottom quartiles of each subset. The seven overlapping pathways are shown in a frame, and immune-related pathways are highlighted in red. **b** Immune scores based on ESTIMATE evaluations in basal tumors (METABRIC $n = 328$; TCGA $n = 190$), BCK-high (METABRIC $n = 376$; TCGA $n = 202$) and low (METABRIC $n = 376$; TCGA $n = 202$) subsets of ER+ tumors. Box plots span the upper quartile (upper limit), median (center) and lower quartile (lower limit). Whiskers extend a maximum of 1.5× IQR. Mann–Whitney *U*-test (two-sided) was used for comparison. **c** Lymphocytes and leukocyte fractions[67] comparisons among TCGA basal subtype tumors ($n = 161$), ER+ BCK-high ($n = 163$) and low ($n = 179$) tumors. Box plots span the upper quartile (upper limit), median (center) and lower quartile (lower limit). Whiskers extend a maximum of 1.5× IQR. Mann–Whitney *U*-test (two-sided) was applied. **d** Kaplan–Meier plots showing the disease-specific survival (DSS) (METABRIC) and overall survival (OS) (TCGA) comparing patients with ER+ BCKs high vs. low tumors. Censored patients were labeled in cross symbols. Log-rank test (two-sided) was used and hazard ratio with 95% CI were labeled. **e** Immune scores based on ESTIMATE evaluations in *ESR1* mutant ($n = 7$) and WT metastatic ($n = 44$) lesions. Box plots span the upper quartile (upper limit), median (center) and lower quartile (lower limit). Whiskers extend a maximum of 1.5× IQR. Mann–Whitney *U*-test (two-sided) was used. **f** Dot plot showing the enrichment level alterations of immune cell subtypes in *ESR1* mutant metastatic lesions using Davoli[68] and Tamborero[69] signatures between *ESR1* mutant ($n = 7$) and WT ($n = 44$) tumors. Significantly increased immune cell subtypes in *ESR1* mutant tumors were labeled in red ($p < 0.05$). Source data are provided as a Source Data file for **b**–**f**.

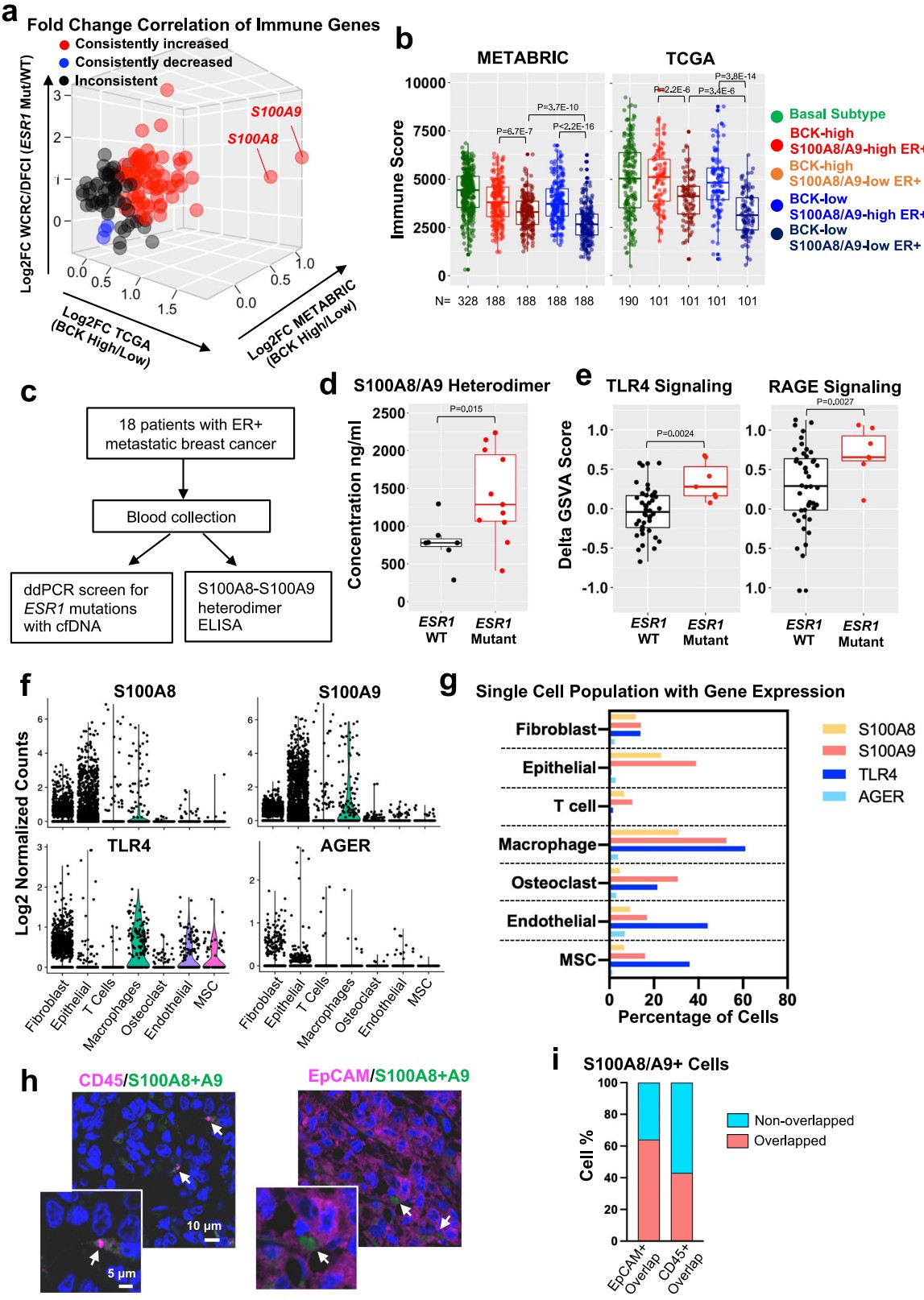

cancer[86], opening up the possibility that BCK-associated immune alterations might contribute to the inferior outcome of patients with *ESR1* mutant breast cancer. The undoubtedly complex role of immune infiltrates in ER+ breast cancer, particularly in the setting of *ESR1* mutant disease, requires further thorough investigation. Intriguingly, our analysis revealed the immune checkpoint gene *VISTA* to be upregulated in *ESR1* mutant tumors.

*VISTA* occupies a unique position as a candidate for cancer immune therapy. Several pre-clinical studies have showed *VISTA* blockade could efficiently enhance immune activation and abrogated tumor immune escape[87]. In addition, two VISTA antagonists, JNJ-61610588 and CA-170 are currently under evaluation in clinical trials[71], suggesting the necessity of elucidating the role of *VISTA* in *ESR1* mutant breast cancer using comprehensive

**Fig. 6 Immune activation in *ESR1* mutant tumors is associated with S100A8/A9-TLR4 paracrine crosstalk. a** Expressional fold changes of immune genes from ESTIMATE (*n* = 141)[73] comparing ER+ BCK-high vs. low tumors (TCGA and METABRIC) and *ESR1* WT/mutant tumors. Consistently increase, decreased or inconsistent genes in are highlighted in red, blue, and black. **b** BCK-high and low quantiles of ER+ tumors were further divided by the mean of *S100A8* and *S100A9*. Immune scores were compared across all four subsets (*n* = 188 and 101 per group of METABRIC and TCGA) together with basal tumors (*n* = 328 METABRIC and *n* = 190 TCGA). Box plots span the upper quartile (upper limit), median (center) and lower quartile (lower limit). Whiskers extend a maximum of 1.5× IQR. Mann–Whitney *U*-test (two-sided) was used. **c** Graphical presentation of experimental strategy of **d**. **d** S100A8/9 heterodimer concentrations in plasma from patients with *ESR1* WT (*n* = 7) and mutant (*n* = 11) metastatic breast cancer. Box plots span the upper quartile (upper limit), median (center) and lower quartile (lower limit). Whiskers extend a maximum of 1.5× IQR. Mann–Whitney *U*-test (two-sided) was utilized. This experiment was done once. **e** TLR4 and RAGE signature enrichments between *ESR1* mutant (*n* = 7) and WT (*n* = 44) tumors. Delta GSVA score was calculated by subtracting the scores of primary tumors from the matched metastatic tumors. Box plots span the upper quartile (upper limit), median (center), and lower quartile (lower limit). Whiskers extend a maximum of 1.5× IQR. Mann–Whitney *U*-test (two-sided) was performed. **f** Violin plots showing expression of four genes by log2 normalized counts in different cell subtypes using single-cell RNA-seq data from two ER+ bone metastases. **g** Percent of cells expressing *S100A8*, *S100A9*, *TLR4*, and *AGER*, using single cell RNA seq data shown in **f**. **h** Immunofluorescent staining of S100A8/A9 with CD45 or EpCAM in a Y537S *ESR1* mutant liver metastasis. Double positive cells are pointed out and magnified. This experiment was done once. **i** Percentage of S100A8/A9+ cells overlapped with EpCAM+ or CD45+ cells. Data were quantified based on six representative regions of the section. Source data are provided as a Source Data file for **a**, **b**, **d**–**g**, **i**.

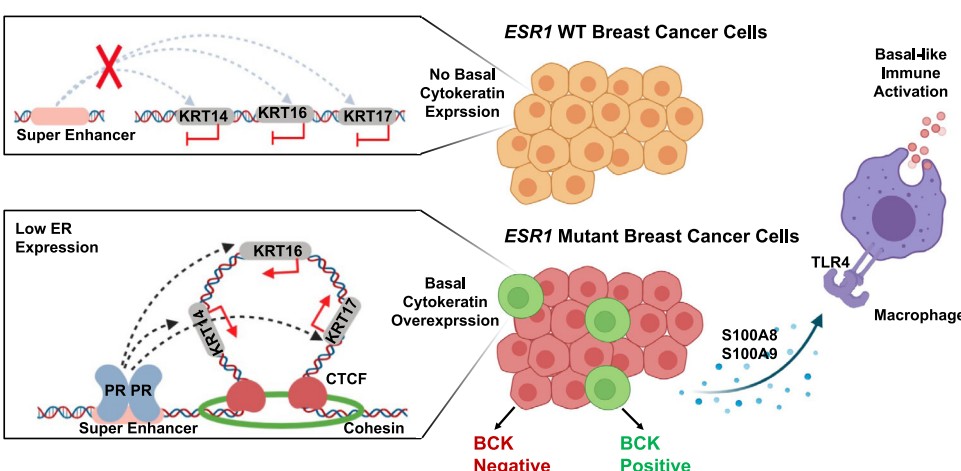

**Fig. 7 Graphical presentation of proposed mechanisms and relevance of basal cytokeratin induction in *ESR1* mutant breast cancer.** *ESR1* WT cells exhibit low basal cytokeratin expression with baseline insulated neighborhood prevalence spanning *KRT14/16/17* loci. In contrast, a minor subpopulation of *ESR1* mutant cells exhibit strong basal cytokeratin expression, due to PR activated enhancer at the *KRT14/16/17* gene locus-spanning insulated neighborhoods. Increased expression of basal cytokeratin is associated with immune activation in *ESR1* mutant tumor similar to that seen in basal tumors, at least in part mediated via enhanced S100A8/A9-TLR4 paracrine crosstalk between epithelial and stromal cells, including macrophages. Figure is generated using BioRender.

immune competent models. Together, these data imply opportunities for immune therapies for patients with ER mutant tumors that should be analyzed further.

We and others[26,27,39] previously identified genes that have altered expression in *ESR1* mutant cells but are not E2 regulated in WT cells. Here, all six BCK belong to this group of gain-of-function target genes according to ER ChIP-seq. Technically, our ChIP-seq exhibited fewer peaks compared other reported models[25,26,39]. This could be possibly owning to our more stringent hormone deprivation protocol or the brand/batch of charcoal-stripped serum utilized in the experiment, which could largely influence the sensitivity of readout detection[88]. Nevertheless, our ChIP-seq data showed considerable overlapped proportion with our data sets and all these ER ChIP-seq supported that BCKs are not regulated as a result of ligand-mimicking nor de novo transactivation by mutant ER, and their expression is strongly and negatively correlated with ER levels. A similar correlation was also observed with P4-induced CK5+ luminal breast cancer cells displaying low ER and PR levels[60]. One possible explanation is that ER, regardless of its liganded status or genotype, serves as a direct epigenetic suppressor that represses BCK expression to maintain luminal identity. For example, it has been shown that ER silences basal, EMT and stem cell related genes by

recruiting pivotal methyl-transferases like EZH2 and DNMTs to reshape the DNA and histone methylation landscape[89]. This could also explain why BCKs mRNAs are increased upon ER knockdown despite PR downregulation: ER and PR control BCK expression through two independent routes. PR only triggers BCKs expression via transcriptional activation on the basis of unique insulated neighborhoods in *ESR1* mutant cells, whereas ER serves as a higher-level epigenetic suppressor in both *ESR1* WT and mutant cells. Knockdown of ER removes the epigenetic repression and allows chromatin accessibility for multiple transcriptional activators, which turns on BCKs expression regardless of PR downregulation. We can not exclude an important role of additional *ESR1* mutant-specific transcriptional regulator beyond PR, although our data lead us to exclude GR. More studies are required to further elucidate the regulatory network between ER, PR, and BCKs. Given bi-directional interactions between tumor and stromal cells in BCK regulation, it will be important to perform future studies in improved model systems such as those recently described for analysis of complex regulation of CK14 expression and function[90].

Assessment of BCK expression revealed that a 50-fold increase in mRNA was reflected in only ~1% cells being positive for BCK protein. This finding is consistent with a previous study showing

that P4 stimulation of breast cancer cells caused a 100-fold induction of CK5 promoter activation ultimately translating to 1–10% of cells positive for CK5 protein[60]. In addition, discordance between mRNA and protein of CK7 and CK14 in breast cancer tissue has been documented[91]. It is possible that BCK protein translation in luminal cells is aberrant, resulting in poorly localized or transported protein, consistent with our detection of BCK protein foci rather than the broad distribution pattern over full cytoskeleton similar to what has been previously reported for example for formation of CK17 foci. The discordance in mRNA and protein expression may be due to the cell heterogeneity, with individual cells having high mRNA and protein compared to the negative population, potentially due to heterogenous expression of miRNAs regulating BCK expression[92]. These BCKs positive cells might be pre-selected by multiple genetic and epigenetic cues including but not limited to low ER expression and chromatin loop formation as identified in our study. The discordance between mRNA and protein expression may also help to explain differences in prognosis using mRNA expression profiling like in our study vs IHC in previous studies[93,94].

We provide evidence to support BCK as emerging biomarkers of ESR1 mutant breast cancer and its prognosis, yet their direct functional impact remains ambiguous. CK14 positive cells typically lead collective invasion across major subtypes of breast cancer cells[95], and this is in line with previously identified enhanced cell migration in ESR1 mutant cells[96]. In addition, as previously described, CK5 positive luminal cells acquire stem-like properties and chemotherapy resistance[48,60]. Importantly, we found several other consistently increased basal marker genes such as interferon-alpha inducible protein 27 (IFI27). Previous studies have reported a role of IFI27 in regulating innate immunity in breast cancer[97] and cisplatin resistance in gastric cancer[36]. Thus, the "basal-ness" shift might confer several broad functional alterations to ESR1 mutant tumors.

We identified a PR-orchestrated insulated neighborhoods at the KRT14/16/17 genomic locus in ESR1 mutant cells, and we propose that the simultaneous generation of a de novo CTCF loop and ER ligand-independent PR overexpression is necessary for KRT14/16/17 in ESR1 mutant cells. Intriguingly, knockdown of CTCF selectively increased KRT14/16/17 mRNA levels whereas knockdown of PR blocked their induction in ESR1 mutant cells. This unexpected discrepancy may highlight that CTCF binding may simultaneously serve as a transcriptional insulator to restrict KRT14/16/17 in an inactive compartment[54,98]. Importantly, data indicates that CTCF knockdown alone is not sufficient to eliminate insulated neighborhoods but instead promotes the formation of new chromatin interactions that alter gene expression[99]. We also unexpectedly found that both PR agonist P4 and PR antagonist RU486 elevated BCK expression, which was inconsistent with previous reported findings where P4 and RU486 exhibited opposite effects in regulating CK5[60]. Given RU486 is well-characterized for its partial agonism, it is possible that ESR1 mutant cells uniquely express a particular strong PR coactivator that confers the partial agonism of RU486 in this context. Another possibility is that RU486 alternatively stimulates other nuclear receptors such as glucocorticoid[93,100] or potentially even androgen receptor[101] to reprogram BCKs expression. The reversed PR pharmacological response in ESR1 mutant cells is intriguing and warrants future investigation.

Our study discovered a unique aspect of ESR1 mutant cells and addressed the underlying mechanisms as well as its clinical relevance, albeit with some remaining limitations, such as limited numbers of clinical samples due to inherent difficulties of obtaining metastatic tissues. The enhanced immune infiltration requires additional validation by TIL counting on ESR1 mutant tumor sections. Confirmation and studies in in vivo models

should be included into future studies. Our preliminary analysis in a ESR1 Y541S (mouse ortholog of Y537S mutation) knock-in mouse model showed overexpression of BCK at RNA and protein level in mammary tumors[102]. And finally, the in silico prediction of enhanced CTCF-driven chromatin loop at the basal cytokeratin gene locus requires confirmation by orthogonal approaches, such as chromosome conformation capture. Nonetheless, our study serves as a robust pre-clinical report uncovering mechanistic insights into ESR1 mutations and their roles in conferring basal-like feature to ER+ breast cancer and implicates the immune therapeutic vulnerabilities to this subset of patients.

## Methods

The authors confirm that this study complies with all relevant ethical regulations. The use of human specimens has been approved by the University of Pittsburgh Human Research Protection Office (HRPO).

**Human tissue and blood studies.** Fifty-one paired primary matched metastatic samples were from DFCI ($n = 15$) and our Women's Cancer Research Center (WCRC) ($n = 36$) cohorts as previously reported[103,104]. For all WCRC metastatic samples, ESR1 mutations status were called from RNA-sequencing. For bone/brain/GI metastatic lesions, ESR1 mutations status were additionally examined using droplet digital PCR for Y537S/C/N and D538G mutations in ESR1 LBD region as previously reported[105]. For DFCI cohort, ESR1 mutations were all called from matched whole exome sequencing[106].

Deidentified samples were obtained from the Pitt Biospecimen Core (PBC) with patients consenting to either the Breast Disease Research Repository (BDRR) (04-162) or to HCC-16-082 (STUDY19060376). For the study of de-identified blood, samples were also obtained from PBC with patients consenting to BDRR (04-162), and the samples were accessed using STUDY19040404 or STUDY19070357 approved by the University of Pittsburgh Institutional Review Board.

**ESR1 mutation detection using droplet digital PCR.** Hotspot ESR1 mutation identification procedure was previously described by us[107]. In brief, blood samples were collected in EDTA tubes (BD, #367856) and cfDNA was isolated from plasma samples using Qiagen circulating nucleic acid kit (#55114). ESR1 ligand binding domain was pre-amplified in cfDNA and the products were subjected to droplet digital PCR detection with Y537S/C/N and D538G probes. Droplets were analyzed using BioRad droplet reader and QuantaSoft software (BioRad, Version 1.7)

**Cell culture.** Establishments of rAAV-edited MCF7 (Park lab)[27], CRISPR-Cas9-edited MCF7 (Gertz[39] and Ali[25] lab) and CRISPR-Cas9-edited T47D cells[27] were reported previously. Individual clones were maintained in DMEM, supplemented with 10% FBS, 100 μg/mL penicillin and 100 mg/mL streptomycin, at 37 °C in a humidified incubator with 5% CO$_2$. Mutation allele frequencies were confirmed using ddPCR. For all experiments, hormone deprivation was performed unless stated otherwise, cells were maintained in phenol-red-free IMEM (Gibco, A10488) with 5% charcoal-stripped serum (CSS, Gemini, #100-119), twice a day for three days. For genome-edited models with multiple clones, individual clones with the same genotypes were equally pooled for subsequent experiments.

Generation of BCK4 cells has been previously reported[108]. MDA-MB-468 (HTB-132), MDA-MB-134-VI (HTB-132), MDA-MB-330 (HTB-127), ZR75-1 (CRL-1500) were obtained from ATCC. Cell lines were maintained in the following media (Life Technologies) with 10% FBS: MDA-MB-468 in DMEM, MDA-MB-134-VI and MDA-MB-330 in 1:1 DMEM: L-15, ZR75-1 in RPMI and BCK4 in MEM with nonessential amino acids (Thermo Fisher, #11140050) and insulin (Sigma-Aldrich, #91077C).

MCF7 (Park lab)/T47D (Oesterreich lab) ESR1 WT and mutant cells (April, 2016), MDA-MB-468 (May, 2020), T47D (February, 2017), ZR75-1 (October, 2016), MDA-MB-134-VI (January, 2020), MDA-MB-330 (January, 2020), and BCK4 (May, 2020) were authenticated at the University of Arizona Genetics Core using autosomal STR profiling with specific time indicated. MCF7 and Y537S ESR1 mutant cell model from Ali lab were authenticated by LGC Standards in March, 2016. MCF7 ESR1 WT and mutant cell lines from Gertz lab were authenticated as matching the correct parental origin using STR marker analysis in August, 2017.

**Generation of MCF7 stable overexpression ESR1 mutation cell model.** To generate ESR1 mutant overexpression cell models, ESR1 WT and mutant plasmids in pcDNA3.1 backbone were obtained from Addgene (ESR1-HA-WT #49498; ESR1-HA-Y537S #49499; ESR1-HA-D538G #49500, Empty vector #V790-20). MCF7 parental cells (ATCC, HTB-22) maintained in 10% FBS DMEM were transfected with each of the plasmid and subjected to 500 μg/mL G418 (Thermo Fisher, #10131035) selection for 3 weeks. G418-containing medium was changed every 3 days during the selection process. Overexpression of WT and mutant ER was further examined by immunoblot and ddPCR in pooled clones and used for

further experiments. MCF7 parental cell line was authenticated at the University of Arizona Genetics Core using autosomal STR profiling in July, 2017.

**Reagents**. siRNA against *ESR1*(L-003401), *PGR* (L-003433), *CTCF* (L-020165), *NR3C1* (L-003424) and non-targeting scrambled control (D-001820-01) were purchased from Horizon Discovery. Progesterone (P1030) and RU486 (m8046) were obtained from Sigma-Aldrich.

**Immunofluorescent staining**. MCF7 cells were hormone deprived and seeded on coverslips. After desired treatments, cells were fixed with 4% paraformaldehyde and blocked with 3% BSA solution plus 0.1% tritonX-100. Primary antibody against CK8 (Abcam, ab53280, 1:100), CK5 (Abcam, ab52635, 1:100), CK16 (Abcam, ab76416, 1:100) and CK17 (Cell Signaling Technology, #4543, 1:100) was applied to stain the cells. For counterstaining, CK16 (Santa Cruz, sc-53255, 1:100) and ER (Leica, PA0151, 1:100) mouse monoclonal antibodies were used to combine with above-mentioned rabbit CK5/16/17 antibodies. Secondary Alexa Fluor 488 or 546-conjugated antibodies (Thermo Scientific, A11008 & A11018, 1:200) and Hoechst (Thermo Scientific, #62249) were used following primary antibody incubation. Coverslips were mounted and images were taken using fluorescence microscope (Olympus, CZX16) under objective of 20× using cellSens software (Olympus Version 1.4). CK5/16/17 positivity quantification was performed by dividing cells with full cytoskeleton CK expression to total cell numbers of each image. For ER counterstaining quantification, ER signal intensity was quantified using ImageJ for each CK positive cell and five proximal CK negative cells.

For staining on tissues, samples were fixed in 10% buffered formalin. Tissue was processed, paraffin embedded, and cut into 5 μm sections. After high-temperature antigen retrieval in citrate buffer (pH 6.0). Sections were permeabilized with 0.1% Triton-X100 in PBS and blocked with 10% normal goat serum for 1 h. Sections were stained with primary antibodies specific for CK5 (rabbit, 1:200, ab75869, Abcam), CK17 (rabbit, 1:100, ab109725, Abcam), PR (M3569, mouse, 1:100, Agilent) EpCAM (rabbit, 1:100, ab71916, Abcam), CD45 (rabbit, 1:200, ab10558, Abcam), or S100A8/9 (mouse, 1:200, NBP1-60157, Novus Biologicals, Littleton, CO, USA) for 2 h at RT. Sections were stained with secondary antibodies Alexa Fluor goat-anti-mouse 488 (1:200, A-11001, Life Technologies, Rockville, MD, USA) and Alexa Fluor goat-anti-rabbit 647 (1:200, A-21245, Life Technologies) for 1 h at RT. Sections were counterstained with Hoechst dye (1:2000 in PBS, Life Technologies). Slides were mounted using Fluoro-Gel Mounting Medium with Tris buffer (1798510, Electron Microscopy Sciences, Hatfield, PA, USA). Slides were imaged using an Olympus IX83 fluorescent microscope or a Nikon A1R confocal microscope. The HCI-013EI PDX model was obtained from Dr. Alana Welm at Univesity of Utah, Huntsman Cancer Institute with Material Transfer Agreement (MTA002948).

**Immunohistochemistry staining**. Tissue sections were processed as above and were stained with antibodies specific for CK5 (rabbit, 1:200, ab75869, Abcam, Cambridge, UK) and CK17 (rabbit, 1:100, ab109725, Abcam) for 2 h at RT. Sections were blocked using HRP Blocking Reagent (Abcam) EnVision+/HRP Visualization (Agilent, Santa Clara, CA, USA) and DAB substrate kit (Agilent) were used to visualize staining. Sections were counterstained with hematoxylin and mounted with Permount Mounting Medium (Fisher Scientific, Waltham, MA, USA). Representative photographs were taken under a light microscope at ×20 magnification.

**Quantitative real-time polymerase chain reaction (qRT-PCR)**. Different cell models were seeded into 6-well plate with 120,000 cells per well using biological triplicates. After the respective treatments, RNAs were extracted using Qiagen RNeasy Kit, and cDNAs were synthesized using PrimeScript RT Master Mix (Takara Bio, #RR036). qRT-PCR reactions were performed with SybrGreen Supermix (BioRad, #1726275) and data were recorded using CFX Manager software (BioRad Version 3.1), and the ΔΔCt method was used to analyze relative mRNA fold changes and RPLP0 levels were measured as the internal control. All primer sequences are shown in Supplementary Table 7.

**Immunoblotting**. Cells were lysed with RIPA buffer plus protease and phosphatase cocktail (Thermo Scientific, #78442) and sonicated. Protein concentration of each sample was determined by Pierce BCA assay kit (ThermoFisher, #23225). Forty microgram (ER, HA, and PR) or 120 μg (CK5 and CK16) proteins were loaded onto 10% SDS-PAGE gel, and then transferred onto PVDF (ER, HA, and PR blot) or NC (CK5 and CK16 blot) membrane. The blots were immune-stained with corresponding antibodies. For ER, HA, and PR blots visualization, Licor blot fluorescence scanner was used following IRDye 680LT or 800CW secondary antibodies incubation using Image Studio software (Licor, Version 3.1). For CK5 and CK16 blot, chemiluminescent approach was used following Amersham HRP-linked secondary antibody (Millipore Sigma (GENA934) and Clarity Western ECL substrate (BioRad, #1705061) incubation. Antibodies against ER (#8644, 1:1000), HA-tag (#3724, 1:1000) and PR (#3176, 1:500) were purchased from Cell Signaling. Tubulin antibody was obtained from Sigma (T5168, 1:5000). Antibody against CK5 (ab52635, 1:500) and CK16 (ab76416, 1:500) were from Abcam. Uncropped blots images are provided in the Supplementary Fig. 13.

**S100A8/S100A9 heterodimer ELISA**. Human S100A8/S100A9 heterodimer amounts in human plasma samples were quantified using S100A8/S100A9 heterodimer Quantikine ELISA kit (R&D System, DS8900) following the manufacture protocol. All plasma samples were first diluted in calibration buffer with 1:50 ratio and loaded into antibody-coated plate.

**Cytokine array with human-derived macrophages**. Human-derived monocytes were obtained from a leukopak from a healthy donor. Monocytes were treated with M-CSF (Peprotech, 300-025) for 5 days to differentiate into M0 macrophages. Cells were then treated with medium alone, 100 ng/ml lipopolysaccharide (LPS, Millipore Sigma, L4391) or 10 μg/ml recombinant human S100A8/S100A9 heterodimer protein (R&D Systems, 8226-S8) for 24 h. Seven hundred microliter of cell supernatant was harvested for each sample and centrifuged to remove particles. Supernatants were analyzed with the Proteome Profiler Human Cytokine Array Kit (R&D Systems, ARY005B) following manufacturer protocol. Briefly, membranes were blocked with blocking buffer supplied by the kit. Samples were diluted with assay buffer as described in the manufacturers protocol and incubated overnight with membranes and antibody cocktail. Membranes were washed, incubated with Streptavidin-HRP buffer supplied in the kit and incubated for 30 min at room temperature. Arrays were washed and imaged by chemiluminescence using a BioRad ChemiDoc XRS + molecular imager.

**Chromatin-immunoprecipitation (ChIP) and sequencing analysis**. ChIP was performed as previously described[52]. Briefly, hormone-deprived MCF7 WT and mutant cells were treated with vehicle or 1 nM E2 for 45 min. Chromatin DNA was then extracted from each sample. The immunoprecipitation was performed using CTCF (EMD Millipore, 07-729), ERα (Santa Cruz Biotechnologies, sc543) and rabbit IgG (Santa Cruz Biotechnologies, sc2027) antibodies. 5 μg antibody were used for each sample. For CTCF ChIP, qPCR was employed and fold enrichment method was used to quantify the binding enrichment at the selected sites. All primers used was recorded in Supplementary Table 7. For ER ChIP-seq, pooled DNA samples from individual clones were sent to McGill sequencing core using Illumina Hiseq 2000 Platform.

ChIP-seq reads were aligned to hg38 genome assembly using Bowtie 2.0[109], and peaks were called using MACS2.0 with $p$ value below 10E−5[110]. We used DiffBind package (Version 2.2.12)[111] to perform principle component analysis, identify differentially expressed binding sites and analyze intersection ratios with other data sets. Heatmaps and intensity plots for binding peaks were visualized by EaSeq. Annotation of genes at peak proximity was conducted using ChIPseeker (Version 1.26.0)[112], taking the promoter region as +/− 3000 bp of the transcriptional start site (TSS) and 50 kb as peak flank distance.

**RNA sequencing analysis**. RNA sequencing data sets were analyzed using R version 3.6.1. Log2 (TPM + 1) values were used for the RNA-seq of Oesterreich *ESR1* mutant cell models and TMM normalized Log2(CPM + 1) values were used for Gertz RNA-seq data. TCGA reads were reprocessed using Salmon v0.14.1[113] and Log2 (TPM + 1) values were used. For the METABRIC data set, normalized probe intensity values were obtained from Synapse. For genes with multiple probes, probes with the highest inter-quartile range (IQR) were selected to represent the gene. For pan-breast cancer cell line transcriptomic clustering, 97 breast cancer cell line RNA-seq data were reprocessed using Salmon and merged from three studies[34–36], batch effects were removed using "removeBatchEffect" function of "limma" package (Version 3.46.0). Gene set variation analysis was performed using "GSVA" package (Version 1.38.0)[115]. Survival comparisons were processed using "survminer" packages (Version 0.4.8) using Cox Proportional-Hazards model and log-rank test. Data visualizations were performed using "ggpubr (Version 0.4.0)", "VennDiagram (Version 1.6.2)"[116] and "plot3D (Version 1.0)". PAM50 subtype prediction was performed using "genefu (Version 2.26.0)" package[117] using the 97 breast cancer cell line panel as reference.

**Single-cell RNA-sequencing analysis**. Two bilateral bone metastases (BoMs) were collected from a patient initially diagnosed with ER+ primary breast cancer, and immediately dissociated into single cells using tumor dissociation kit from Miltenyi Biotech (130095929) following manufacturer's protocol. Red blood cell lysis (Qiagen158904) and dead cell removal (Miltenyi Biotech 130090101) were performed according to the manual.

Raw counts were mapped to human genome assembly (version GRCh38) using cellranger count function, and the mapped count matrix was imported into Seurat (v 3.1.4) for further analysis. Genes with detected expression in less than 20 cells, as well as cells expressing less than 300 genes or more than 8000 genes, or containing more than 45% mitochondrial genes were removed, resulting in 10,056 cells for downstream analysis. Mitochondrial genes were regressed out before principle component analysis, and a shared nearest neighbor optimization based clustering method was used for identifying cell clusters. Cell type of each cluster was assigned by the expression of canonical cell markers, and cell signatures derived from single cells RNA sequencing data of defined cell types collected in PanglaoDB database. Log normalized counts values of *S100A8, S100A9, TLR4*, and *AGER* were compared between different cell types.

**Tumor mutation burden (TMB) analysis**. TMB calculation was performed as previous described[118]. Briefly, TCGA mutation annotation files from 982 patients were downloaded from FireBrowse and mutation subtypes were summarized using "maftool" package (Version 2.6.0)[119]. Mutations subtypes were classified into truncated (nonsense, frame-shift deletion, frame-shift insertion, splice-site) and non-truncated mutations (missense, in-frame deletion, in-frame insertion, non-stop). TMB was calculated as 2× Truncating mutation numbers + non-truncating mutation numbers.

**Generation of gene sets**. For Sorlie et al., the original set of intrinsic genes were downloaded from Stanford Genomics Breast Cancer Consortium (http://genome-www.stanford.edu/breast_cancer/). Four hundred and fifty three genes were annotated from 553 probes. Expression of these 453 genes were examined in 33 luminal and 39 basal breast cancer cell lines. Significantly higher (FDR < 0.01) intrinsic genes in basal or luminal cells were called as basal ($n = 75$) or luminal ($n = 68$) markers in Sorlie gene sets. For the TCGA and METABRIC gene set, differentially expressed genes were called between basal and luminal A or basal and luminal B ER+ tumors using raw counts or normalized probe intensities by DESeq2 (Version 1.34.0) or limma (Version 3.46.0). The top 200 increased genes of these two comparisons were further intersected. Overlapped DE genes in basal and luminal tumors were called as TCGA and METABRIC gene sets. For CTCF gene signature establishment, a previous RNA-seq data set on MCF7 cells with or without CTCF knockdown was downloaded and analyzed[53], top 100 down-regulated genes with CTCF knockdown were used as the CTCF gene signature. All the gene signatures used in this study are provide in Supplementary Data 1.

**Chromatin interaction data analysis**. CTCF ChIA-PET data were downloaded from GSE72816. Chromatin linkages were visualized on 3D genome browser (http://promoter.bx.psu.edu/hi-c/) after processed with ChIA-PET tool (Version 3)[120]. Confident insulated neighborhoods boundaries were defined by the colocalization of CTCF and cohesion complex subunits together with called chromatin interactions. Hi-C data were downloaded from GSE130916, hiC file was visualized using WashU Epigenome Browser.

**Super-enhancer identification**. Super-enhancers were identified from a widely used human super enhancer database-SEdb[121] (http://www.licpathway.net/sedb/index.php) which curates and processes H3K27ac ChIP-seq data sets from publicly available resources and further computes super enhancers using the ROSE pipeline[122]. Specifically, we used the super enhancer information called from a MCF7 H3K27ac ChIP-seq data set from GSE57436[123]. The recognized super enhancer at *KRT14/16/17* region was ranked #25 among all 210 super enhancers.

**Statistical analysis**. All statistical analysis were specified at the corresponding figure legends. For data processing and visualization, Microsoft Excel 2020, Graphpad Prism (Version 8) and R (version 3.6.1) was used.

**Reporting summary**. Further information on research design is available in the Nature Research Reporting Summary linked to this article.

## Data availability
The ER ChIP-seq data from MCF7 *ESR1* mutant cell model and single-cell RNA data set from two bone metastasis generated in this study have been deposited in Gene Expression Omnibus with accession number of GSE125117 and GSE190772. MSigDB curated gene sets were downloaded from GSEA website [http://software.broadinstitute.org/gsea/msigdb/index.jsp]. RNA-seq data and clinical information from TCGA and METABRIC were obtained from the GSE62944 and Synapse software platform under accession number syn1688369, respectively. TCGA biospecimen immune profile data were downloaded from Saltz et al.[67]. TCGA mutation annotation format (MAF) files and methylation data were downloaded from FireBrowse website [http://firebrowse.org/]. The DFCI data set has been described in previous publications[26,124–127]. Complete RNA-seq data from DFCI cohort have been deposited into the database of Genotypes and Phenotypes (dbGaP) under accession number phs001285.v1.p1 and will be available following standard restriction policy of dbGaP. Further questions regarding data access should be sent to Dr. Nikhil Wagle (Nikhil_Wagle@dfci.harvard.edu). RNA-Seq data for the paired primary and metastatic samples in WCRC cohort has been previously generated and reported[18,104,128]. Raw sequencing data cannot be published openly in order to protect participants identities but will be made available upon request by the corresponding author Dr. Steffi Oesterreich (oesterreichs@upmc.edu) under regulatory compliance via a data usage agreement (DUA). Data can be accessed as soon as documents according to required policy have been completed. No further restrictions will be applied. For all WCRC samples, the processed data (transcript counts processed via Salmon) are available at [https://github.com/leeoesterreich]. Processed RNA-seq data (in TMM normalized Log2(CPM + 1)) from both WCRC and DFCI cohorts used in this study (51 pairs of primary-metastatic tumors) and the annotation files can be accessed in R data file format with the corresponding codes from Code Ocean [https://codeocean.com/capsule/2816027/tree/v1]. File names are: DFCI_pairs_DF.Rda; WCRC_pairs_DF.Rda;

DFCI_Key.Rda; WCRC_Key.Rda. All the raw data and scripts are available upon request from the corresponding author. Sources of all public available data sets used in this study are summarized in Supplementary Table 8. The remaining data are available within the Article, Supplementary Information or Source Data file. Source data are provided with this paper.

## Code availability
R script associated with this study was deposited into Code Ocean and can be accessed using the link https://codeocean.com/capsule/2816027/tree/v1.

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

## Acknowledgements

We thank Dr. Peilu Wang for her contribution to earlier *ESR1* mutant-studies in the Lee-Oesterreich group. We thank Corrine Farrell, Jian Chen, Dr. Jagmohan Hooda, Christy Smolak and Dr. Peter Lucas for some technical support of the project. We also thank Dr. Alana Welm from University of Utah for providing the HCI-013EI PDX model. This project used the University of Pittsburgh Pitt Biospecimen Core (PBC), Cytometry Facility, and Tissue and Research Pathology Service (TARPS) supported in part by award NIH grant P30CA047904. The authors would like to thank the patients who contributed samples to the tissue bank as well as all the clinicians and staff for their efforts in collecting tissues. This work was supported by a Susan G. Komen Scholar awards [SAC110021 to A.V.L.]; the National Cancer Institute [R01CA221303 to S.O. (Oesterreich) and P30CA047904]; the Department of Defense [W81XWH1910499 to S.O. (Oesterreich); DOD grant W81XWH1910434 (J.G.)]; Magee-Women's Research Institute and Foundation, Nicole Meloche Foundation, Penguins Alumni Foundation, the Pennsylvania Breast Cancer Coalition and the Shear Family Foundation. S.O. (Oesterreich) and A.V.L. are Hillman Fellows. Z.L. is supported by John S. Lazo Cancer Pharmacology Fellowship. The content is solely the responsibility of the authors and does not necessarily represent the official views of the National Institutes of Health or other Institutes and Foundations.

## Author contributions

Z.L., J.M.A., A.V.L., and S.Oe conceived and designed the study. Z.L., O.M., S.On, C.L., T.C.B., Y.W., A.B., and K.D. designed, performed and analyzed experiments. Z.L., A.B., N.M.P., and K.D. performed bioinformatic analysis. J.M.A., L.M., and M.R. contributed to clinical sample collection and intellectual input. N.W. provided extended RNA-seq data set (DFCI) from clinical specimens and intellectual input. Z.L., A.V.L., S.Oe, T.C.B., D.A.A.V., C.A.S., J.K.R., W.J.M., and J.G. contributed to data interpretation and provided additional intellectual input. L.B., S.A., and J.G. provided additional cell models for this study and intellectual input. Y.F., L.Z., and G.C.T. provided and validated biostatistical approaches of all the analysis. Z.L., A.V.L., and S.Oe developed the figures and the manuscript. All the authors reviewed and agreed with the contents of the manuscript.

## Competing interests

S.Oe and A.V.L. receive research support from AstraZeneca PLC. A.V.L. is employee and consultant with UPMC Enterprises, and member of the Scientific Advisory Board, Stockholder and receives compensation from Ocean Genomics. Tsinghua University paid the stipend of University of Pittsburgh-affiliated foreign scholar Yang Wu from Tsinghua

University. D.A.A.V. is cofounder and stock holder of Novasenta, Potenza, Tizona, Trishula; stock holder of Oncorus, Werewolf, Apeximmune; patents licensed and royalties of Astellas, BMS, Novasenta; scientific advisory board member—Tizona, Werewolf, F-Star, Bicara, Apeximmune; consultant of Astellas, BMS, Almirall, Incyte, G1 Therapeutics; research funding of BMS, Astellas, and Novasenta. The remaining authors declare no competing financial interests.
