## [Peer Review File · Nature Communications]

Reviewers' Comments:

Reviewer #1:

Remarks to the Author:

The manuscript by Li et al describes the enrichment of basal cytokeratins in ESR1 mutant ER+ breast cancers that is mediated through PR associated chromatin reprogramming of the KRT14/16/17 genomic region. The authors also show that increased BCK expression is associated with a good patient prognosis and is associated with higher immune inactivation, and conclude that consideration of immunotherapeutic approaches is warranted.

Major comments

1. The authors first look at the 'luminal-ness' and 'basal-ness' of ESR1 mutant and wild-type cell lines and primary tumors from TCGA. It would also be useful to subtype these cell lines/ clinical samples with PAM50 or equivalent clinical assays. In ESR1 mutant cells, is the association more boarder-line? Why was METABRIC not used here given the larger samples numbers?
2. The authors find that ESR1 mutant cells show subclonal expression of CK5/16/17+ cells. Can the authors also validate this finding in clinical samples using IHC/IF?
3. The authors find a relationship between elevated PR expression and regulation of BCK's via a CTCF-driven chromatin loop in ESR1 mutant MCF7 cell line models. Are the authors able to validate these findings in patient samples?
4. The use pathway enrichment to identify enrichment of immune related pathways in BCK high tumors. From METABRIC and TCGA. Given these datasets are from primary disease, is there any association with ESR1 mutation status here? If not- could the authors speculate whether these patients would be more likely t develop ESR1 mutations upon the course of their disease? Were these survival associations borne out in multivariate analyses?
5. Immune cell type deconvolution analysis identified increased CD8+ cells in ESR1 mutant metastatic tumors? Was this also seen in the H&E's of the tumors? Was this also the case in TCGA/METABRIC cohorts? There is evidence in the literature suggesting that higher immune infiltrate and patterns of immune infiltrate are associated with poorer response to therapy (PMID: 28859291), this should be discussed in the manuscript. What percentage of immune infiltrate did these tumors show? This would be useful to assess to benchmark relative to ER- and ER-/HER2+ tumors.
6. The authors finally identify S100A8/9 expression to be high in ESR1 mutant vs wild-type and BCK high versus low tumors and cell line models and use single cell sequencing from 1 patient to assess which cell fraction highest expression is seen in. These transcripts are also highly elevated in stress response (i.e when tissues are dissociated for single cell RNA-seq), thus the authors should show that these cells are not dying in the single cell data analysis. The cut-offs used seem very high for cells with expression of mRNA reads (45%). In addition, the authors should validate this in additional patients as this is n=1, using IHC/IF to validate the cell populations that express high levels. i.e. is this signal really from the epithelial cells, or highest in associated immune cells? Is expression of S100A8/9 independently associated with prognosis?
7. It would also be useful for the authors to assess PDL-1 distribution in ESR1 mutant versus wild-type tumors if the conclusion is that these patients may benefit from immunotherapeutic approaches.

Minor

- Please make the single cell data available in a public repository not just on the University portal
- Details on the single cell-seq dissociation methodology and analysis should be in the main methods.

Reviewer #2:

Remarks to the Author:

Plasticity is an interesting area in breast cancer. This manuscript has found that ESR1 LBD hotspot mutations including Y537S and D538G gained basal-ness using 4 gene signatures delivered from published datasets. This finding has been validated by breast cancer metastatic samples harbor ESR1 mutations. . Integrating RNA-seq results from cell lines authors found that BCKs are the

predominant basal like markers enriched in mutant cell lines. Next, authors performed ChIPseq analysis and showed that most upregulated basal genes were not directly regulated by ER suggesting that these genes are secondary effect of mutant. PR has been shown by the authors to regulate the super enhancer region around CK14,16,17 genes. PR antagonist RU486 was able to activate CK16 in mutant cell lines using IF. Finally, authors tried to link basal like markers to immune activation. In clinical datasets, both BCK and ER high patients were correlated with higher immune score. S100A8 and S100A9 were the top over expressed immune genes in mutants and they were validated in mutant patient blood samples. This is a very interesting manuscript which showed how plasticity involved in ESR1 mutant models. Authors used high throughput sequencing to identify regions of interest. In addition, authors have validated their pre-clinical finding in clinical samples. Here are some comments:

1. "Basal-ness" and over expression of S100A8, S100A9 in mutants have been published in reference paper 73 which made this manuscript lack of novelty. However, authors have deeply studies these two areas.
2. In figure 3 and 4, PR as an important regulator of the BCKs transcription identified through ChIPseq is not convincing. When ESR1 knockdown was performed, PR will be significantly abolished in mutant cells, thus Low PR should reduce the expression of BCKs. however, in the paper figure 3D, BCKs were enhanced which suggested that PR may not be the major player in the BCK overexpression. GR could also bind within the same response element, and GR maybe also upregulated in the mutant cells.
3. Authors mentioned immune activation was associated with mutant in Figure 5. However, this conclusion is missing some key experiments to fully support the conclusion. For example, using human PBMC co-culture with mutant cells or WT cells and look at the T cell activation. S100A8 and S100A9 could play an important role in MDSC regulation which could play a role for immune suppressive phenotype.
4. Figure 6 is descriptive; perform K.D. of S100A8 and S100A9 in ER positive immune competent in vivo models would be the best way to end the paper by quantifying infiltrated lymphocyte.

Reviewer #3:

Remarks to the Author:

This manuscript aims to explore the potential role of ESR1 mutations in subtype switching from luminal to basal and its association with increase immune activation. Estrogen receptor alpha (ERα/ESR1) is mutated in 30-40% of endocrine-resistant ER-positive (ER+) breast cancer and its mutation causes ligand-independent growth and increased metastasis in vivo and in vitro. Overall, this manuscript reports their novel findings in ESR1 mutant breast cancer cells through defining a basal cytokeratin induction that works through PR regulated super-enhancer and TAD domain organization. Their studies further supported the concept that the increase in basal-ness of ESR1 mutant breast tumors is associated with immune activation in part facilitated by the paracrine S100A8/A9-TLR4 signaling. The findings are very interesting and will provide the important knowledge to design a new drug strategy in ESR1 mutant tumors through immune therapeutic vulnerabilities. Although the authors have presented extensive data, a number of major concerns given below still need to be addressed to improve the whole manuscript.

1. In Figure 1, the gain of "basal-ness" in ESR1 mutants is very clear. Is there a loss of "lumina-ness" in ESR1 mutants? Can the authors check this possibility and confirm it with some analyses?
2. In the previously published paper (Rinath Jeselsohn, Cancer Cell, 2018), they identified 35, 000 binding sites in Y537S mutant cells and 11,371 sites in D538G mutant cells. However, in this manuscript, the authors only identified 657 binding sites in Y537S and 1,016 in D538G mutant cells. Especially, the authors only identified 12,472 peaks for WT ERα under E2 stimulation condition, which is also much lower than the ERα binding peaks identified by many previous published ERα ChIP-seq data. The authors need to make sure that their ChIP-seq quality is good enough. Otherwise, their claim that no ERα peak identified within 50kb from TSS of BCK genes in ESR1 mutant cells could be weakened by the bad ChIP-seq data created in this manuscript. What

about using the ChIP-seq data from Rinath's paper to confirm some of their important conclusions?

3. The authors observed an increase of BCK genes after the knockdown of ESR1 in MCF7. Together with other data, the authors claimed ER is the negative regulator of BCK genes in ESR1 mutant tumors. The authors further claimed PR as a positive regulator for BCK gene expression. One confusing thing is that estrogen/ER is a well-known positive regulator for PR gene expression in MCF7 and other ER+ breast cancer cells. How to explain the surprising finding of ER and PR relationship in ESR1 mutant cells? More discussion would be appreciated.

4. In Figure 4C and 4D, the authors claimed the TAD domain covering the KRT14/16/17 loci. Besides ChIA-PET, did the authors use any Hi-C based sequencing data to predict the TAD domains? From the size of those CTCF interacting regions, I would like to use "insulated neighborhood" to indicate such regions. Anyway, the better definition details on the TAD and insulated neighborhood term use would be very helpful for the readers.

5. The authors found super-enhancers around KRT14 and KRT16 locus. But the details on how the enhancers were identified were not stated clearly in the paper. Did the authors use H3K27ac or Mediator ChIP-seq data for the super-enhancer identification analyses?

6. Considering the ESR1 mutant is associated with the BCK activation, immune activation, and better survival rate in this manuscript, it seems contradictory to the current knowledge that ER mutant is a bad predictor for patient survival. It will be appreciated if the authors can discuss this more in the Discussion section.

Point-to-point reply to reviewer comments for Nature Communications manuscript (NCOMMS-20-51419A-Z)

We greatly appreciate the in-depth and constructive comments from all three reviewers on our manuscript. Below is a point-to-point reply towards each single point. Reviewers' comments are in italic, our **direct responses are in red** and our **corresponding edits to the manuscript files are in blue** for readability. New and modified figures are included below, and figures labeled with "R" are for Reviewers/Editor only (ie not part of the revised manuscript).

Overall, the reviewers were very excited about our study. Their comments such as "This is a very interesting manuscript which showed how plasticity involved in *ESR1* mutant models." (Reviewer #2) and "The findings are very interesting and will provide the important knowledge to design a new drug strategy in *ESR1* mutant tumors through immune therapeutic vulnerabilities." (Reviewer#3) corroborated the novelty and potential clinical impact of our findings.

Reviewer#1

1. The authors first look at the 'luminal-ness' and 'basal-ness' of *ESR1* mutant and wild-type cell lines and primary tumors from TCGA. It would also be useful to subtype these cell lines/ clinical samples with PAM50 or equivalent clinical assays. In *ESR1* mutant cells, is the association more boarder-line? Why was METABRIC not used here given the larger samples numbers?

RESPONSE:

We thank the reviewer for providing these valuable suggestions. We performed some additional analysis to address the questions.

First, we used R package *genefu* (PMID: 26607490) to predict the PAM50 subtype probabilities of MCF7 *ESR1* WT and mutant cell lines with a merged transcriptomic data set of other 97 breast cancer cell lines as reference. Consistent with previous data, both Y537S and D538G mutant cells

are dominantly classified as basal subtype (>70% basal probability) whereas WT cell are classified as luminal A (new Supplementary Fig. S3A).

We further performed a principal component analysis using the PAM50 gene expression matrix to visualize the clustering status of *ESR1* mutant cells among different subtypes. There is an obvious trend of shift of both *ESR1* mutant

Supplementary Fig S3A. Stacked plot shows probability distribution of five molecular breast cancer subtypes in MCF7 *ESR1* WT and mutant cells. The dominant subtype called by *genefu* is indicated with asterisk symbols.

Supplementary Fig S3B. PCA plot showing the clustering of *ESR1* WT and mutant cells with other 97 characterized breast cancer cell lines based on PAM50 gene expression.

cells towards basal cell population while *ESR1* WT and WT+E2 groups remain at the same cluster with luminal/HER2 cell lines (new Supplementary Fig. S3B).

These additional analyses confirmed our initial finding that *ESR1* mutant cell lines showed increase “basal-ness”.

We also attempted a similar analysis in our patient-matched paired cohort but did not observe significant differences in basal subtype probabilities between *ESR1* WT and mutant metastatic samples (shown here in Fig. R1). As suggested by the reviewer, our analysis showed that the enrichment of basal gene sets in *ESR1* mutant cells/tumors was dependent on a broader range of basal marker genes, while the PAM50 panel only includes a smaller set of representative basal genes which might weaken the power to compute the basal subtype shift. This of course is even more unfavorable due to relatively small sample size of the *ESR1* mutant tumors.

Figure R1
Dot plot representing the basal subtype probabilities of 51 ER+ metastatic tumors from WCRC/DFCI cohort called by geneфу (44 *ESR1* WT and 7 *ESR1* mutant samples). Mann Whitney U test was applied.

Second, as suggested by the reviewer, we generated basal and luminal gene sets using METABRIC cohort following the identical procedure used for TCGA in the original manuscript. Briefly, we computed the differentially expressing genes from comparisons of luminal A vs basal and luminal B vs basal tumors and intersected the top 200 differentially expressed (DE) genes between these two comparisons, which ultimately generated 133 and 157 genes as METABRIC basal and luminal gene sets respectively (new Supplementary Fig. S1E).

Supplementary Fig. S1E. Schematic procedure of generation of METABRIC gene sets. Briefly, differentially expressed genes (FDR<0.01) were called between basal and luminal A or basal and luminal B ER+ tumors using log2 normalized probe intensities of each gene. The top 200 increased genes from these two comparisons were further intersected (Venn diagram). Shared upregulated (n=133) or downregulated (n=157) DE genes between basal-to-LumA and basal-to-LumB comparisons tumors were called as METABRIC gene sets.

Intersection of these two gene sets with the four other pairs we used in the original analysis showed somewhat limited overlaps, although canonical luminal genes (e.g. *FOXA1*, *ESR1*) and basal genes (e.g. *KRT6A*, *KRT16*)

Fig. 1B. Venn diagrams representing the overlap of genes from all five basal (left) and luminal (right) gene sets. Genes overlapping in four gene sets are indicated.

(e.g. *KRT6A*, *KRT16*) were observed in at least 4 of all 5 genes sets (modified Fig. 1B).

We next repeated the GSEA analysis with the METABRIC-derived gene signatures. As shown in modified Fig. 1C-1F, and similar to the data in the original submission, we observed the significant enrichment of basal gene sets in MCF7 *ESR1* mutant cell models and

ESR1 mutant metastasis (modified Fig. 1C and 1E), whereas no consistent differences were discerned for the luminal gene sets (modified Fig. 1D and 1F). In summary, these additional analysis

with METABRIC-derived signatures further strengthened our conclusion that *ESR1* mutant cells and tumors show increased “basal-ness”.

Figure 1C and 1D. Dot plots showing GSEA score of the five pairs of basal (C) and luminal (D) gene sets enrichment in MCF7 genome-edited cell models. Scores from luminal and basal breast cancer cell lines were used as positive controls. Dunnett’s test was used to compare with WT-vehicle set within each gene set. (* $p < 0.05$, ** $p < 0.01$)

Figure 1E and 1F. Box plots representing basal (E) and luminal (F) gene set enrichments in intra-patient matched paired primary-metastatic samples. Delta GSEA score for each sample was calculated by subtracting the scores of primary tumors from the matched metastatic tumors. Mann-Whitney U test was performed to compare the Delta GSEA scores between WT (N=44) or *ESR1* mutation-harboring (N=7) paired tumors. (* $p < 0.05$, ** $p < 0.01$)

EDITS:

1. Page 3, line 39

We replaced “four” with “five” since we added the METABRIC gene set into the analysis.

2. Page 7, line 110

We replaced “four” with “five” since we added the METABRIC gene set into the analysis.

3. Page 7, line 113

We replaced “two” with “three” since we added the METABRIC gene set into the analysis.

4. Page 7, line 116

We added “METABRIC” into the description and refer Supplementary Fig. S1E.

5. Page 7, line 120

We replaced “3 out of 4” with “4 out of 5” since we added the METABRIC gene set into the analysis.

6. Page 7, line 121

We replaced “four” with “five” since we added the METABRIC gene set into the analysis.

7. Page 7, line 129

We added the following sentence to highlight the consistent result from PAM50-based prediction.

‘This was further corroborated by PAM50-based analysis, where we found MCF7 *ESR1* mutant cells were predominantly called as basal subtype with above 70% probability (Supplementary Fig. S3A) and exhibited gene expressional similarities to basal breast cancer cell lines (Supplementary Fig. S3B).’

8. Page 9, line 152

We replaced “four” with “five” since we added the METABRIC gene set into the analysis and update the union gene number “N=742”.

9. Page 12, line 223

We replaced “four” with “five” since we added the METABRIC gene set into the analysis and update the union gene number “N=742”.

10. Method section

We added the method of PAM50 prediction in the “RNA sequencing analysis” section.

We added the description of METABRIC gene signatures generation and the PAM50 subtype prediction in the methods section.

11. Figure 1/Figure 2A/Supplementary Fig. S1, S2, S3, S4, S7

We updated all the subpanels by adding the corresponding information and results from METABRIC gene sets.

2. The authors find that *ESR1* mutant cells show subclonal expression of CK5/16/17+ cells. Can the authors also validate this finding in clinical samples using IHC/IF?

RESPONSE:

It remains a challenge to obtain access to sufficient metastatic samples with specific mutations of interest for staining of candidate genes. To identify *ESR1* mutant metastatic tumors with sufficient material to cut sections for IHC/IF staining, we screened frozen metastatic tissues (n=13) using droplet digital PCR for four *ESR1* hotspot mutations (Y537S, Y537C, Y537N and D538G). We identified a liver metastasis harboring a Y537S mutation with 27.1% mutant allele fraction in genomic DNA and 79.1% fraction in cDNA (new Supplementary Fig. S4F).

We then cut and stained sections from this metastatic sample with validated antibodies against CK5 and CK17 using an MCF10A cell pellet as positive control. Confirming our observation using *ESR1*

mutant cell lines, we observed heterogenous expression of CK5 and CK17 in subclonal pattern in the

ESR1 mutant liver metastasis by both IF and IHC (new Fig. 2E).

Fig. 2E Immunofluorescent (left panel) and immunohistochemistry (right panel) staining of CK5 and CK17 on sections from MCF10A (positive control) and a Y537S *ESR1* mutant liver metastasis tissue. Images were taken under 10X (IF) or 20X (IHC) magnification. Subclones with CK5 or CK17 expression were further magnified and highlighted with white arrow.

EDITS:

1. Page 10, Line 191

We added the following sentence to describe clinical sample confirmation.

“Importantly, the heterogenous expression of CK5 and CK17 was confirmed in an ER+ liver metastatic lesion harboring an Y537S mutation (Fig. 2E and Supplementary Fig. S4F).”

2. Method section

We added a description of the method for CK5/CK7 IHC/IF and the sources of the liver metastatic sample.

3. Supplementary Fig. S4

We added the ddPCR validation of Y537S in the liver metastasis as Supplementary Fig. S4F.

4. Figure. 2

We added the CK5 and CK17 IF and IHC staining as main figure 2E.

3. *The authors find a relationship between elevated PR expression and regulation of BCK's via a CTCF-driven chromatin loop in ESR1 mutant MCF7 cell line models. Are the authors able to validate these findings in patient samples?*

RESPONSE:

It is challenging to validate a relationship between elevated PR expression and regulation of BCK's via a CTCF-driven chromatin loop in clinical samples. However, we did attempt to address the reviewer's concern at least in part while at the same time aiming to identify models that could be used for further mechanistic studies. Briefly, we utilized a PDX model with an Y537S *ESR1* mutation (HCl-013EI, PMID: 34717714) (obtained via MTA from Dr Alana Welm at the Huntsman Cancer Institute, Salt Lake City, Utah), and performed co-staining of CK17 and PR. We identified tumor cells that were double positive for CK17 and PR (new Supplementary Fig. S10G). Additional studies that we feel are

outside the scope of this manuscript will utilize this model to validate the role of CTCF-driven chromatin loop in PR's regulation of BCKs.

EDITS:

1. Page 15, Line 305

We added the following sentence to describe this result.

“Identification of double positive (CK17+ and PR+) cells in a Y537S *ESR1* mutant patient-derived xenograft tumor (HCI-013EI)⁶⁴ provides further support for regulation of BCK by PR (Supplementary Fig. S10G).”

2. Method section

We edited the methods to include CK17 and PR IF, and acknowledged the source of HCI-013EI PDX obtained via MTA from Dr. Welm at HCI.

4. Supplementary Fig. S10

We added the PR/CK17 co-staining data as Supplementary Fig. S10G.

4. The use pathway enrichment to identify enrichment of immune related pathways in BCK high tumors. From METABRIC and TCGA. Given these datasets are from primary disease, is there any association with *ESR1* mutation status here? If not- could the authors speculate whether these patients would be more likely to develop *ESR1* mutations upon the course of their disease?

RESPONSE:

To our knowledge, neither cohort has released molecular profiling data from metastases paired with the primary tumors described in these large cohorts. Thus, it is not possible to compare *ESR1* mutation frequencies in metastatic lesions between BCK high and low primary tumors in TCGA nor METABRIC.

However, to try to address the reviewer's comment we utilized our in-house paired primary tumor- metastasis cohort in this study, albeit the number is limited (n=51). Specifically, we compared the enrichment levels of the six BCKs in two groups of primary tumors, whose matched metastatic tumors were either *ESR1* WT or *ESR1* mutant. This analysis showed that primary tumors that subsequently showed *ESR1* mutations in the paired metastasis did not show differential BCK enrichment (Fig. R2), in contrast there is a trend towards lower BCK expression in primary tumors with paired *ESR1* mutant metastases. These data suggest that clones with high BCK expression in primary tumors may not be the same clones harboring *ESR1* mutations in the subsequent metastatic lesions after progression.

Were these survival associations borne out in multivariate analyses?

As for the second question, we performed a multivariate analysis using Cox Proportional Hazard Models in both the TCGA and METABRIC cohorts. Our analysis merged BCK enrichment levels with other clinical features that i) are known to affect survival, and ii) are available for the cohorts. The result showed a significant association between BCK enrichment and favorable prognosis in TCGA, and a similar trend was seen in METABRIC (new Supplementary Fig. S12F), confirming that patients whose primary tumors have high BCK expression have improved survival.

EDITS:

1. Page 17, line 349

We modified the following sentence to describe the multivariable survival analysis.

“Intriguingly, patients with BCK-high ER+ tumors experience improved outcomes in both uni- and multivariable analysis (Fig. 5D and Supplementary Fig. S12F).”

2. Supplementary Fig. S12

We added the forest plot of multivariable survival analysis as Supplementary Fig. S12F.

5. Immune cell type deconvolution analysis identified increased CD8+ cells in ESR1 mutant metastatic tumors? Was this also seen in the H&E's of the tumors?

RESPONSE:

To address this question, we contacted van Geelen et al. at the Peter MacCallum Cancer Center (Australia) who recently performed TIL counting in parallel to targeted panel DNA sequencing on a large series of breast cancer metastases (PMID:32811538). They kindly shared the data from 93 ER+

metastatic tumors with information on *ESR1* genotype and TIL percentages. We compared the TIL% between *ESR1* WT (n=83) and mutant tumor (n=10) specimens and did not detect a significant difference.

There are several possible explanations for this data in relation to our deconvolution analysis presented in Figure 5F. *First*, our results in metastatic tumors were corrected to the immune scores of matched primary tumors to eliminate the influence from patient-specific immune background variations, whereas no such correction was performed in this cohort. *Second*, 4 of the 10 *ESR1* mutant metastatic sites of this cohort were lymph node metastasis, which our cohort does not harbor. It is plausible that the T cell percentage from lymph nodes, as a source of immune cells, may not truly reflect the tumor-related immune infiltrate and the remaining distant metastatic solid tumors may not provide sufficient statistic power due to low number. And *finally*, it is likely that only specific immune cell subtypes are increased in the metastases but not the general T cell infiltration. We think that a multi-center study is necessary to obtain a sufficiently large cohort of *ESR1* mutant metastases that can be analyzed using a comprehensive panel covering a wide range of immune cell subtypes. We hope that the Reviewer and Editor agree that such study is outside the scope of this current manuscript. We have however included a sentence into the Discussion towards the need for such study to validate and expand our results.

Was this also the case in TCGA/METABRIC cohorts?

Unfortunately, such analysis is not possible since TCGA or METABRIC cohorts are limited to primary treatment-naïve tumors which do not harbor *ESR1* mutations.

We could however compare CD8 T cell signatures between BCK-high and low tumors to validate our GSVA analysis (new Supplementary Fig. S12G). Similar with the observations in metastatic tumors, both Davoli and Tamboraro CD8 T cell signatures are increased in BCK high ER+ tumors (new Supplementary Fig S12G) albeit still lower than in the basal subtype tumors.

There is evidence in the literature suggesting that higher immune infiltrate and patterns of immune infiltrate are associated with poorer response to therapy (PMID: 28859291), this should be discussed in the manuscript.

We thank the reviewer for this comment. We have included the reference and discussed this finding in our revised manuscript.

What percentage of immune infiltrate did these tumors show? This would be useful to assess to benchmark relative to ER- and ER-/HER2+ tumors.

This is an excellent question and relates to the comment above regarding the amount of CD8 T cell infiltration. In collaboration with the Vignali lab (PITT), we are currently performing a comprehensive analysis of immune cell infiltration in a large set of ER+ and ER- tumors (>100) using FACS, Vectra staining and single cell sequencing. This analysis showed less infiltration of B cells, T cells, and T

regs but an increase in macrophages in ER+ compared to ER- tumors. We believe that a comprehensive description of the immune infiltrate in these tumors compared to ER- tumors is outside the scope of our current manuscript, but we could refer to our other study in the discussion if the reviewer and Editor believe that is helpful.

Motivated by the comment, however, we have performed a bioinformatic analysis to test whether BCK levels could stratify immune response in ER-/HER2+ or TNBC tumors. As shown in new Supplementary Fig. S12D, we found no difference in immune scores between BCK high and low in HER2+ tumors. BCK high TNBC even showed a lower immune score than the BCK low counterpart in the METABRIC cohort. Taken together, this analysis suggests that the positive association between high BCK expression and immune activation is specific for ER+ tumors.

Supplementary Fig. S12D. Dot plots showing the immune score based on ESTIMATE evaluations in BCK-high and low subsets of TNBC and ER-/HER2+ tumors in TCGA and METABRIC cohorts. Specific sample numbers are labelled below each subpanel. BCK high and low subsets were separated based on median of GSVA score. Mann Whitney U test was used (** p<0.01)

EDITS:

1. Page 17, line 342

We modified the sentence below to refer to the negative results identified in ER-/HER2+ and TNBC tumors in TCGA and METABRIC.

“An orthogonal approach - bioinformatic evaluation using ESTIMATE⁶⁶ - confirmed the unique enhancement of immune activation in BCK-high vs BCK-low ER+ tumors which is not seen in ER-/HER2+ or TNBC subtypes (Supplementary Fig. S12D), albeit overall it is still lower than in basal tumors (Fig. 5B).”

2. Page 17, line 355

We added the following sentence as a confirmation of CD8 T cell signature increase in BCK high ER+ primary tumors.

“The higher CD8+ T cell scores were also observed in BCK-high primary tumors in TCGA and METABRIC (Supplementary Fig. S12G).”

3. Page 22, line 448

We added the following sentence to emphasize the need for additional immune characterization in *ESR1* WT and mutant tumors including analysis of spatial heterogeneity into the revised Discussion.

“Additional multi-center studies are necessary to comprehensively characterize immune infiltration in *ESR1* mutant tumors, including the analysis of spatial heterogeneity, as a recent study demonstrates that spatial clustering of immune cells (Immune Spatial Score) is linked to poor recurrence-free survival in ER+ breast cancers⁸⁵. Immune activation by S100A8/A9 may reshape the architecture of cancer-immune ecosystem.”

4. Supplementary Fig. S12

We added the ESTIMATE score comparison in ER-/HER2+ and TNBC subsets as Supplementary Fig. S12D, and the CD8 T cell signature comparison between BCK high and low ER+ tumors as Supplementary Fig. S12G.

6. The authors finally identify *S100A8/9* expression to be high in *ESR1* mutant vs wild-type and *BCK* high versus low tumors and cell line models and use single cell sequencing from 1 patient to assess which cell fraction highest expression is seen in. These transcripts are also highly elevated in stress response (i.e. when tissues are dissociated for single cell RNA-seq), thus the authors should show that these cells are not dying in the single cell data analysis. The cut-offs used seem very high for cells with expression of mRNA reads (45%).

RESPONSE:

As for the first question regarding to the potential cell death in our single cell RNA-seq, we have carefully measured the cell viability using trypan blue staining after tissue dissociation, and confirmed approximately 80%-90% viability, before proceeding to the next step. Given the validated cell viability, the selection of 45% mitochondrial count ratio for our downstream analysis allowed sufficient complexity and numbers of each cell subtype.

To address the reviewer’s concern, we re-analyzed our data using 20% cutoff for mitochondrial count ratio filtration. As shown in Figure R3, despite lower remaining cell numbers, the gene expressional pattern of *S100A8/9*, *TLR4* and *AGER* were very similar to that under 45% mitochondrial count ratio cutoff (Fig. 6F in the original manuscript), where the ligands *S100A8/9* were predominantly expressed in epithelial and the *TLR4* receptor was mainly expressed in stroma. Thus, we believe the mitochondrial ratio we select for data filtration does not influence our conclusion.

Fig R3. Violin plots showing *S100A8*, *S100A9*, *TLR4* and *AGER* expression by log2 normalized counts in different cell subtypes using single-cell RNA-seq data from two bone metastases from a patient with ER+ breast cancer. Cells with mitochondrial counts ratio below 20% were pre-selected for this analysis.

In addition, the authors should validate this in additional patients as this is n=1, using IHC/IF to validate the cell populations that express high levels. i.e. is this signal really from the epithelial cells, or highest in associated immune cells?

Fig 6H. Immunofluorescent counter staining of *S100A8/9* with CD45 (left panel) or EpCAM (right panel) in a Y537S *ESR1* mutant liver metastasis lesion. Cells show dual fluorescence are pointed out with white arrows and magnified.

Fig 6I. Stacked bar plot showing percentage of *S100A8/9*+ cells overlapped with EpCAM+ or CD45+ cells. Data were quantified based on cell counting from six representative regions of the section.

To validate the specific cell subtype expressing *S100A8/9* in ER+ metastatic lesions, we performed dual-color immunofluorescent staining for *S100A8/9* with CD45 or EpCAM in an *ESR1* mutant liver metastatic sample (described in point #2 above).

Consistent with our single-cell RNA-seq analysis, *S100A8/9* was identified in both CD45+ and CD45- cells. Further, counter staining with EpCAM confirmed the expression in a subset of tumor epithelial cells (new Fig. 6H). Quantification of the data showed a greater overlap of

S100A8/A9+ cells with EpCAM+ than CD45+ cells (new Fig. 6I), providing further support for the hypothesis of paracrine signaling identified in our single-cell transcriptome analysis.

Is expression of S100A8/9 independently associated with prognosis?

Supplementary Fig. S12K. Kaplan-Meier plots showing the disease-specific survival (DSS) (METABRIC) and overall survival (OS) (TCGA) comparing patients with ER+ S100A8/A9 high vs low tumors. BCKs high and low were defined by the upper and bottom quartiles of each subset. Censored patients were labelled in cross symbols. Log rank test was used and hazard ratio with 95% CI were labelled.

This question was addressed using data from patients with ER+ breast cancers in TCGA and METABRIC (new Supplementary Fig. S12K). High S100A8/A9 expression was not associated with improved outcomes but rather showed poor prognosis in the METABRIC cohort. (Of note, the clinical data is more reliable in the METABRIC compared to the TCGA cohort).

As we mentioned in the Discussion, despite the fact that

the majority of studies showing an immune activation role of S100A8/A9 in the tumor microenvironment, other studies (e.g. PMID: 31620141) uncovered immune suppressive effects of S100A8/A9 via activation of myeloid-derived suppressor cells (MDSC). This is not unexpected, reflecting complex systems biology, where it is likely that high S100A8/A9 expression triggers more immune repressive than activating effects and is thus associated with poor prognosis. In contrast, BCK high tumors are associated with a diverse range of immune pathways activation such as TNF- α , IL2-STAT5 and IL6-STAT3 as revealed in Fig. 5A of our original manuscript. Thus, we propose that S100A8/A9-associated immune activation is context-dependent and more evident in the context of BCK high tumors. Further investigations on the mechanistic interplay are warranted.

EDITS:

1. Page 19, line 393

We added the following sentence to describe our additional validation of S100A8/A9 expression pattern in clinical specimen.

“This was confirmed by immunofluorescent staining in the Y537S liver metastatic tissue, where we found S100A8/A9+ cells overlapped with approximately 60% and 40% EpCAM+ and CD45+ cells respectively (Fig. 6H and 6I), validating that epithelial cancer cells are a prevalent source for S100A8/A9 production in the ecosystem.”

2. Page 18, line 371

We edited the manuscript to include the data from the survival analysis.

“As expected, S100A8-A9 expression correlated positively with immune scores in ER+ tumors (Fig. 6B), however, S100A8/A9 expression did not associate with improved survival, suggesting a more complex role in this context (Supplementary Fig. S12K).”

3. Figure 6

We added the S100A8/A9 and EpCAM/CD45 counter staining data as the new main figure 6H and 6I.

4. Supplementary Fig. S12

We added the Kaplan-Meier plot of S100A8/A9 in TCGA and METABRIC as Supplementary Fig. S12K.

7. It would also be useful for the authors to assess PDL-1 distribution in *ESR1* mutant versus wild-type tumors if the conclusion is that these patients may benefit from immunotherapeutic approaches.

RESPONSE:

Following the reviewer's suggestion, we compared expression of PD-L1 and a number of other key immune checkpoint genes between *ESR1* WT and mutant metastatic tumors in our paired-met cohort (new Supplementary Fig. S12H). While no differential expression of the majority of immune checkpoint genes were observed (including PD-L1), we found a significant increase of *VISTA* in *ESR1* mutant tumors. Previous studies have identified *VISTA* as a multi-lineage immune checkpoint that serves as key suppressor of T cell-associated response prompting immune evasion in different types of human cancers (PMID: 32600443). Importantly, *VISTA* occupies a unique position as a candidate for cancer immune therapy. Several pre-clinical studies have showed *VISTA* blockade could efficiently enhance immune activation and abrogated tumor immune escape (e.g. PMID: 30382166). In addition, two *VISTA* antagonists, JNJ-61610588 and CA-170 are currently in clinical trials (PMID: 32554470). We expect that this new data will provide additional motivation and rationale to test immunotherapeutic approaches in immune competent models hopefully leading to trials in patients with *ESR1* mutant tumors.

EDITS:

1. Page 17, line 357

We added the sentence below to describe the finding of high *VISTA* expression in *ESR1* mutant tumor.

“In addition, immune checkpoint expression analysis revealed higher expression of *VISTA* in *ESR1* mutant tumors (Supplementary Fig. S12H), which has been characterized as a key suppressor of T cell-associated immune response in human cancer⁷⁰ and can be pharmacologically targeted⁷¹.”

2. Page 23, line 460

We added the following sentence in to discussion.

“Intriguingly, our analysis revealed the immune checkpoint gene *VISTA* to be up-regulated in *ESR1* mutant tumors. *VISTA* occupies a unique position as a candidate for cancer immune therapy. Several

pre-clinical studies have showed VISTA blockade could efficiently enhance immune activation and abrogated tumor immune escape⁸⁷. In addition, two VISTA antagonists, JNJ-61610588 and CA-170 are currently under evaluation in clinical trials⁷¹, suggesting the necessity of elucidating the role of VISTA in *ESR1* mutant breast cancer using comprehensive immune competent models.”

3. Supplementary Fig. S12

We added the box plot of immune checkpoint expression comparisons between *ESR1* WT and mutant tumors as Supplementary Fig. S12H.

Minor comments from Reviewer#1

- Please make the single cell data available in a public repository not just on the University portal

RESPONSE:

Absolutely. The single cell RNA-seq data has been deposited into GEO with the accession number GSE190772.

EDITS:

1. Page 35, line 721

We added the GEO deposition information in the Data Availability section.

“ER ChIP-seq data from MCF7 *ESR1* mutant cell model and single-cell RNA data set from two bone metastasis were deposited in Gene Expression Omnibus with accession number of GSE125117 and GSE190772. Single-cell RNA-seq data will be made public upon publication of this study.”

2. Supplementary Table S10

We added the single cell RNA-seq data set source information into the table.

- Details on the single cell-seq dissociation methodology and analysis should be in the main methods.

RESPONSE:

Following the reviewer’s suggestion, we have included the single cell RNA-seq dissociation methods to the Material and Methods section.

EDITS:

Page 32, line 661

We moved the detailed single-cell RNA-seq procedure and analysis description into the main methods.

“Single-cell RNA-sequencing analysis

Two bilateral bone metastases (BoMs) were collected from a patient initially diagnosed with ER+ primary breast cancer, and immediately dissociated into single cells using tumor dissociation kit from Miltenyi Biotech (130095929) following manufacturer’s protocol. Red blood cell lysis (Qiagen158904) and dead cell removal (Miltenyi Biotech 130090101) were performed according to the manual.

Raw counts were mapped to human genome assembly (version GRCh38) using cellranger count function, and the mapped count matrix was imported into Seurat (v 3.1.4) for further analysis. Genes with detected expression in less than 20 cells, as well as cells expressing less than 300 genes or more than 8,000 genes, or containing more than 45% mitochondrial genes were removed, resulting in 10,056 cells for downstream analysis. Mitochondrial genes were regressed out before principle component analysis, and a shared nearest neighbor optimization based clustering method was used

for identifying cell clusters. Cell type of each cluster was assigned by the expression of canonical cell markers, and cell signatures derived from single cells RNA sequencing data of defined cell types collected in PanglaoDB database. Log normalized counts values of S100A8, S100A9, TLR4 and AGER were compared between different cell types.”

Reviewer#2

1. “Basal-ness” and over expression of *S100A8*, *S100A9* in mutants have been published in reference paper 73 which made this manuscript lack of novelty. However, authors have deeply studies these two areas.

RESPONSE:

We thank the reviewer for appreciating the depth of our work. We were aware of the very recent publication by Gu et al. (PMID: 33323970) which pointed out the “basal-ness” and “S100A8/A9” expression in *ESR1* mutant breast cancer and we cited their work accordingly. However, in addition to the depths of our work, there are several novelties we would like to highlight based on our results.

First, we used four different luminal-basal gene set pairs (and now this has increase to five as per Reviewer#1’s comments) and observed the highly consistent basal marker enrichment in two different cell models with two *ESR1* mutant variants (Y537S and D538G). In addition, our conclusion was confirmed in *ESR1* mutant clinical specimens.

Second, our study goes way beyond this as we expanded clinical and mechanistic analysis on two genes from 141 immune-related gene signature as top candidates differentially expressed between *ESR1* WT and mutant tumors and confirmed our finding using patient plasma samples. In addition, our analysis also uncovered the potential functional impact of these two genes in altering the immune response.

EDITS:

No further edits have been made in response to this comment.

2. In figure 3 and 4, *PR* as an important regulator of the *BCKs* transcription identified through *ChIPseq* is not convincing. When *ESR1* knockdown was performed, *PR* will be significantly abolished in mutant cells, thus Low *PR* should reduce the expression of *BCKs*. however, in the paper figure 3D, *BCKs* were enhanced which suggested that *PR* may not be the major player in the *BCK* overexpression.

GR could also bind within the same response element, and *GR* maybe also upregulated in the mutant cells.

RESPONSE:

We agree with the reviewer that first, these results don’t easily come together, and second, that *GR* is and additional candidate gene which might mediate some of the observed effects. To address this concern, we have clarified our hypothesis on ER-PR-BCK interaction and tested the role of *GR* (described below).

We propose that ER and PR might play repressive and activating roles on *BCK* expression, respectively, through two independent mechanisms. Our data showed that knockdown of ER strongly induced *BCKs* expression in all cell types in the absence of ER binding sites. This result, together with previous reports (PMID: 28108626), suggest that ER, regardless of its genotype, is likely an epigenetic repressor that controls chromatin accessibility at a wide range of genomic regions. On the other hand, knockdown of PR uniquely abrogated *BCK* expression in *ESR1* mutant cells, and specific PR binding sites were detected at the chromatin loop regions, suggesting PR as an *ESR1* mutant specific transcriptional activator. Taken together, we propose *BCK* gene expression regulation by ER and PR regulation at two different levels:

a. In the presence of ER, the *BCK* regions are epigenetically silenced in the majority of cells. Only in some clones with relatively low ER expression, a moderate level of PR cooperates with insulated neighborhoods in *ESR1* mutant cells and hence triggers *BCK* expression by transcriptional activation.

Notably, owing to the ligand-independent activation of mutant ER, we expect the clones with low ER expression still exhibit modest PR expression. This is also shown in our BCK-PR counter-staining experiment in *ESR1* mutant PDX tumors. (See Supplementary Fig. S10G and reply to Reviewer#1 Point#3).

b. In the absence of ER (e.g. ER knockdown), the epigenetic repression effects are eliminated and the corresponding chromatin regions can be largely accessed by multiple transcriptional activators, which turns on BCK expression despite PR downregulation.

As pointed out by the reviewer, it is possible that other *ESR1* mutant-specific transcriptional regulators other than PR are involved, including GR. We performed additional bioinformatic and wet bench experiments to test this hypothesis:

First, we explored GR (*NR3C1*) expression in RNA-seq data from cell models. In the MCF7 *ESR1* mutant cell model we used in this study, we found a moderate yet significant repression of *NR3C1* after estrogen stimulation in *ESR1* WT cells, and a decrease of *NR3C1* level in both Y537S and D538G mutant cells (new Supplementary Fig. S11A), which doesn't support the concept that GR plays a role in BCK expression. However, it was plausible that GR binding is increased in the mutant cells, despite lower GR expression. We thus examined BCK mRNA expression using qRT-PCR after 7 days knockdown of GR in MCF7 *ESR1* WT and mutant cell models. We observed that GR knockdown increased BCK expression levels in *ESR1* mutant cells (except for *KRT17*). This effect was very similar to our observation after ER knockdown, suggesting GR might also serve as a transcriptional or epigenetic repressor for BCKs (new Supplementary Fig. S11B).

Supplementary Fig. S11:

A. Bar plot showing expression of *NR3C1* in MCF7 *ESR1* WT and mutant cells. Data were extracted from RNA-seq using log₂(TPM+1) values with four biological replicates. Dunnett test was used to compare each group versus WT vehicle group. (** p < 0.01).

B) qRT-PCR measurement of *ESR1*, *KRT5/6A/6B/14/16/17* mRNA levels in MCF7 WT and *ESR1* mutant cells with *NR3C1* siRNA knockdown for 7 days. mRNA fold change normalized to WT cells; *RPLP0* levels were measured as internal control. Each bar represents mean ± SD with three biological replicates. Data shown are representative from three independent experiments. Student's t-test was used to compare the gene expression between scramble and knockdown groups. (* p < 0.05, ** p < 0.01)

C). Genomic track illustrating the CTCF/cohesion complex binding at *KRT14/16/17* and *KRT5/6A/6B* proximal genomic region in MCF7 cells. CTCF and RAD21 ChIP-seq were downloaded from ENCODE. STAG1 and SMC1A ChIP-seq data were from GEO (GSE25021 and GSE76893). GR ChIP-seq with dexamethasone treatment was downloaded from GSE72249. Y-axis represents signal intensity of each track.

Analysis of GR binding using publicly available GR ChIP-seq data set of MCF7 cells (PMID: 27062924) identified *KRT* GR binding peaks overlapping with PR binding sites at the *KRT14/16/17* but not *KRT5/6A/6B* loci (new Supplementary Fig.S11C), suggesting GR might play a role in transcriptional repression of *KRT14/16/17*. Collectively, our additional experiments suggest that GR is potentially

mediating repression of BCK expression but unlikely to play a role in the observed activation. Additional future studies utilizing genome-wide tools such as CRISPR/Cas9-screen could help to identify additional factors involved in BCK induction in *ESR1* mutant cells.

EDITS:

1. Page 16, line 321

We added the following paragraph to describe the examination of GR.

“Since glucocorticoid receptor (GR, *NR3C1*) shares similar response motif with PR, we tested whether GR could also activate BCKs expression in *ESR1* mutant cells. Unlike the substantial overexpression of PR, GR expression was moderately repressed in *ESR1* mutant MCF7 cells (Supplementary Fig. S11A). Knockdown of GR increased expression of BCKS (except *KRT17*) in both *ESR1* WT and mutant cells (Supplementary Fig. S11B), and GR binding was identified at the super enhancer region at *KRT14/16/17* loci (Supplementary Fig. S11C). These data suggest that although GR can bind to regulatory regions in keratin genes, it is unlikely to play a causative role in BCK induction observed in *ESR1* mutant cells.”

2. Page 24, line 487

We expanded the Discussion to clarify our hypothesis on BCK gene regulation by ER and PR:

“This could also explain why BCKs mRNAs are increased upon ER knockdown despite PR downregulation: ER and PR control BCK expression through two independent routes. PR only triggers BCKs expression via transcriptional activation on the basis of unique insulated neighborhoods in *ESR1* mutant cells, whereas ER serves as a higher-level epigenetic suppressor in both *ESR1* WT and mutant cells. Knockdown of ER removes the epigenetic repression and allows chromatin accessibility for multiple transcriptional activators, which turns on BCKs expression regardless of PR down-regulation. We cannot exclude an important role of additional *ESR1* mutant-specific transcriptional regulator beyond PR, although our data lead us to exclude GR.”

3. *Authors mentioned immune activation was associated with mutant in Figure 5. However, this conclusion is missing some key experiments to fully support the conclusion. For example, using human PBMC co-culture with mutant cells or WT cells and look at the T cell activation. S100A8 and S100A9 could play an important role in MDSC regulation which could play a role for immune suppressive phenotype.*

RESPONSE:

This is an excellent suggestion, and we performed additional experiments to address this question. Since our single-cell RNA-seq analysis revealed a more prevalent crosstalk between epithelial cells and macrophages, we sought to test if increased S100A8/A9 expression in *ESR1* mutant cells triggers macrophage activation. As a first step, we tested whether S100A8/A9 treatment could induce cytokine production from macrophages, reflecting activation of the cells.

Briefly, human-derived monocytes were differentiated to M0 macrophages for 5 days with CSF-1 and then treated with 10 µg/ml recombinant S100A8/A9 proteins or 100ng/ml lipopolysaccharide as a positive control for TLR4 signaling activation for 24 hours. Supernatants were then collected and subjected to cytokine array (R&D System, ARY005B) covering 36 cytokines (Fig. R4).

While LPS activated macrophages and increased secretion of pro-inflammatory cytokines such as IL-6, IL-10 and TNF- α , we did not observe any changes induced by S100A8/A9 treatment. These data suggest that either S100A8/A9 does not trigger macrophage activation as measured by release of cytokines, or (more likely given extensive literature) this *in vitro* system might not be suited to test the role of S100A8/A9. Indeed, our preliminary *in vivo* experiment (shown in Fig. R6 below) showed that pharmacological blockade of S100A8/A9 signaling suppressed overall immune cell abundances in ER+ mouse tumors but did not influence MDSC levels. Thus, the ideal experiment system to test the question raised by the reviewer is to compare the immune landscape in *ESR1* WT and mutant tumors in immune competent models (see #4 below).

Figure R4. Images of cytokine array results using human-derived monocytes under control, LPS and S100A8/A9 treatments for 24 hours. Negative and positive control spots were labelled and representative increased cytokines were highlighted.

EDITS:

Currently, the revised manuscript does not contain any edits in response to this comment. However, we would be delighted to include a few sentences on negative data described above if the reviewer and editor prefer us to comment on this,

4. Figure 6 is descriptive; perform K.D. of S100A8 and S100A9 in ER positive immune competent *in vivo* models would be the best way to end the paper by quantifying infiltrated lymphocyte.

RESPONSE:

We absolutely agree with the reviewer - regulating S100A8/A9 in ER+ immune competent *in vivo* models is the best way to quantify infiltrating lymphocytes. The SSM3 ER+ cell line generated from Stat1 knockout mice (PMID: 27391074) is one of the very few suitable models, and we decided to use this model to perform a preliminary study address the reviewer’s concerns.

qRT-PCR showed considerable S100A8/A9 mRNA expression (Ct value <25) in SSM3-derived primary tumors. H&E-based pathological evaluation confirmed immune infiltration in the tumors (Fig. R5). We therefore conducted an *in vivo* experiment using this cell model.

Two million cells were injected into the inguinal mammary fat pads of ten 129s6/SvEv immune competent mice with 0.5mg bees wax estradiol pellet embedded in each mouse. When tumors reached the volume of approximately 500mm³ (48-50 days of growth), we injected 5mg/kg Paquinimod (a specific S100A8/A9 inhibitor) i.p. once a day for 5 consecutive days. DMSO treatment was given to a control group. At the end of the treatment, mice were sacrificed, and the tumors were harvested. Unfortunately, we had 2 unexpected deaths in the control group and 1 death in the treatment group, resulting in only 3 and 4 tumors from the control and the treatment groups, respectively, available for the immune cell abundance quantification using flow cytometry (Fig. R6A).

Figure R5. Representative H&E images depicting the immune infiltration identified in SSM3 cell-derived primary mammary tumor.

First, and in support of our hypothesis, viable CD45+ immune cells were decreased upon inhibition of S100A8/A9 in these tumors (Fig. R6B, average percentage of 20% in control tumors versus 10% in Paquinimod-treated tumors) although this effect did not reach significance (p=0.067). Further immune

cell subtype profiling among the viable CD45+ cells failed to identify any significant differences with the exception of an increase in regulatory T cells (Tregs) upon S100A8/A9 blockade (Fig. R6C).

Figure R6A. Schematic overview of pilot S100A8/A9 blockade in vivo experiment using SSM3 syngeneic mouse cell line.

Figure R6B. Bar plot showing the percentage of viable CD45+ cells in tumor with or without Paquinimod treatment using flow cytometry quantification after tumor digestion (Control, n=3, Paquinimod, n=4). Student's t test was used.

Figure R6C. Bar plot showing the percentage of different immune cell subtypes among all the viable CD45+ cells in tumors with or without Paquinimod treatment. Student's t test was applied to each individual immune cell type.

While the results from this preliminary experiment are in support of our other data collectively suggesting that increased S100A8/A9 mediates immune infiltration in ER+ breast cancer, the death of 2 and 1 mice in the control and treatment groups, respectively, and overall limited cell viability limits the ability to draw solid and rigor conclusion from this experiment. We can repeat the experiment, however, we feel that over the last year we have spent significant amount of time and resources on the revisions, and we have already asked for one extension to generate additional data. We therefore hope that the Reviewer and Editor are satisfied with this preliminary data which provides some confidence in our overall message of the manuscript.

EDITS:

No edits have been made under this comment.

Reviewer#3

1. In Figure 1, the gain of “basal-ness” in *ESR1* mutants is very clear. Is there a loss of “luminal-ness” in *ESR1* mutants? Can the authors check this possibility and confirm it with some analyses?

RESPONSE:

This is an excellent question which we had addressed in our original manuscript. In all analyses in the manuscript, we always analyzed the enrichment of “luminal-ness” in parallel to that of “basal-ness” using the four different luminal-basal gene signatures (now five in response to Reviewer#1’s suggestion).

Briefly, we did not identify any consistent change of “luminal-ness” in *ESR1* mutant cell models and clinical samples. The results can be seen in Fig. 1D and 1F of the original manuscript. These data suggest that *ESR1* mutations might unidirectionally reprogram “basal-ness” without a major effect seen in “luminal-ness.”

EDITS:

No edits have been made under this comment.

2. In the previously published paper (Rinath Jeselsohn, *Cancer Cell*, 2018), they identified 35, 000 binding sites in Y537S mutant cells and 11,371 sites in D538G mutant cells. However, in this manuscript, the authors only identified 657 binding sites in Y537S and 1,016 in D538G mutant cells. Especially, the authors only identified 12,472 peaks for WT ERα under E2 stimulation condition, which is also much lower than the ERα binding peaks identified by many previous published ERα ChIP-seq data. The authors need to make sure that their ChIP-seq quality is good enough. Otherwise, their claim that no ERα peak identified within 50kb from TSS of BCK genes in *ESR1* mutant cells could be weakened by the bad ChIP-seq data created in this manuscript.

RESPONSE:

This is an important point, and we appreciate the reviewer raising it. We think that our ChIP-seq data is reliable, and of sufficient quality to detect peaks in BCK genes if they were there. The reason for fewer binding site is likely technical, and specifically is related to more intense hormone deprivation. Briefly, we compared our stripping procedure with methods from other published ChIP-seq studies, and concluded that our hormone deprivation procedure is more intense. As described in the Methods Section, we washed the cells in 5% charcoal-stripped serum (CSS) twice per day for three consecutive days, whereas others used a less harsh protocol (e.g. 72 hours incubation without washing in Harrod et al (PMID: 31106278); two medium changes within 5 days in Arneson et al. (PMID:33184109)). It is possible that such intense hormone deprivation will deplete weaker ER bindings resulting in lower overall number of ER peaks. Secondly, our previous report (PMID: 27459541) showed that different brands and batches of CSS could influence the results of estrogen response studies. A sensitive technology like ChIP-seq is likely to be affected by differences in CSS. The CSS we used for this experiment was carefully pre-tested with in vitro cell growth assay to ensure no residual cell growth under hormone deprivation. It is possible that CSS used by groups allows some residual ER activation.

Nevertheless, it is important to note that there is significant overlap in the ER binding sites between our data and that from other publicly available ChIP-seq data from *ESR1* WT and mutant models. We compared our data with that from four other models: MCF7 CRISPR/Cas9 model from Harrod et al. (PMID: 31106278), MCF7 Dox-inducible and TALEN genome-edited *ESR1* mutant cell model from Jeselsohn et al. (PMID: 29438694) and another MCF7 CRISPR/Cas9 model from Arneson et al. (PMID:33184109). As shown in the new Supplementary Fig. S7B, there was an average of 62% and 40% overlap with WT and mutant ER peaks respectively. And finally, we do see overlap at many

canonical binding sites (e.g. *GREB1* shown in Fig 3B), suggesting that the ones we lose are representing weaker binding sites.

Supplementary Fig. S7B. Venn diagrams representing the intersection of ER ChIP-seq used in the study to other three independent data sets from Harrod et al. (Ali model, GSE78286), Jeselsohn et al. (Brown model, GSE94493) and Arneson et al. (Gertz model, GSE148279) ChIP-seq profile from the three different conditions were analyzed including *ESR1* WT+E2, Y537S and D538G mutant cells. Peak overlapping percentages of ChIP-seq from this study were indicated below each diagram.

What about using the ChIP-seq data from Rinath's paper to confirm some of their important conclusions?

Following the reviewer's suggestion, we explored ER binding sites around BCK genes in four additional ER mutant ChIP-seq data sets including the one from Jeselsohn et al. (PMID: 29438694) (new Supplementary Fig. S7D) Consistent with the data presented in the original submission, we did not identify additional gained mutant ER binding sites at ± 50 kb of all six BCKs in any of the data sets. This analysis confirmed that BCKs are not transcriptionally linked to mutant ER genomic binding.

Supplementary Fig. S7D. Genomic track showing called ER binding peaks at *KRT14/16/17* (left panel) and *KRT5/6A/6B* (right panel) loci from four different ER ChIP-seq data sets of MCF7 *ESR1* mutant cells from Harrod et al. (Ali model, GSE78286), Jeselsohn et al. (Brown model, GSE94493) and Arneson et al. (Gertz model, GSE148279).

EDITS:

1. Page 11, line 211

We added the description of the ChIP-seq overlap with other data sets.

“Furthermore, intersection analysis with other reported ChIP-seq data sets of *ESR1* mutant cells revealed considerable intersection ratios (Supplementary Fig. S7B), despite some inter-model variations.”

2. Page 11, line 219

We added the following sentence to depict the confirmation from other ChIP-seq data sets.

“This was further corroborated in four additional ER ChIP-seq data sets in MCF7 *ESR1* mutant cell models (Supplementary Fig. S7D).”

3. Supplementary Figure 7

We added the intersection analysis and genomic track view of other ER ChIP-seq data sets as new Supplementary Fig. S7B and S7D.

3. The authors observed an increase of BCK genes after the knockdown of *ESR1* in MCF7. Together with other data, the authors claimed ER is the negative regulator of BCK genes in *ESR1* mutant tumors. The authors further claimed PR as a positive regulator for BCK gene expression. One confusing thing is that estrogen/ER is a well-known positive regulator for PR gene expression in MCF7 and other ER+ breast cancer cells. How to explain the surprising finding of ER and PR relationship in *ESR1* mutant cells? More discussion would be appreciated.

RESPONSE:

This point has been addressed in detail in our response towards Reviewer#2 point#2.

4. In Figure 4C and 4D, the authors claimed the TAD domain covering the *KRT14/16/17* loci. Besides ChIA-PET, did the authors use any Hi-C based sequencing data to predict the TAD domains?

RESPONSE:

That’s an excellent idea, and following the reviewer’s suggestion, we examined the intra-chromosomal interaction status at *KRT14/16/17* and *KRT5/6A/6B* loci using a previously reported MCF7 Hi-C data set (GSE130916). Data visualization confirmed the substantial chromatin interaction between the two major CTCF binding sites (new Supplementary Fig. S10B, CTCF peak #1 and #5 at *KRT14/16/17* locus identified in our study), whereas no noticeable strong interactions were identified at *KRT5/6A/6B* loci, consistent with our conclusion based on analysis of CTCF ChIA-PET.

Supplementary Fig. S10B. Genomic track illustrating CTCF/Cohesin complex binding and heatmap presentation of chromatin interaction scores at *KRT14/16/17* and *KRT5/6A/6B* loci using a MCF7 Hi-C data set (GSE130916). Each bin represents a 10kb window. The interaction between CTCF peak#1 and #5 at *KRT14/16/17* loci was highlighted with a blue frame.

From the size of those CTCF interacting regions, I would like to use “insulated neighborhood” to indicate such regions. Anyway, the better definition details on the TAD and insulated neighborhood term use would be very helpful for the readers.

We would like to thank the reviewer for pointing out the important nuances of terms used to describe higher order chromatin regions. The median size of a topological associated domain and an insulated neighborhood is 880kb and 190kb respectively (PMID: 30989119, 27863240). The distance between the two interaction sites at *KRT14/16/17* loci reported in our present study is approximately 145kb, which therefore belongs to an insulated neighborhood, as the reviewer stated. We have corrected the terminologies through the entire manuscript.

EDITS:

1. Page 14, line 283

We added the following sentence to describe the Hi-C data confirmation.

“and visualization of a Hi-C data set in MCF7 cell line (Supplementary Fig. S10B).”

2. Page 14, line 276, 278 and 283; Page 26, line 530 and 538; Page 34, line 705

We replaced the term “Topological associated domain” or “TADs” with “insulated neighborhoods”.

3. Page 34, line 707

In Methods section, we added the data source and visualization method for this Hi-C data set.

“Hi-C data were downloaded from GSE130916, hiC file was visualized using WashU Epigenome Browser.”

4. Supplementary Fig. S10

We added the Hi-C data visualization as Supplementary Fig. S10B.

5. The authors found super-enhancers around *KRT14* and *KRT16* locus. But the details on how the enhancers were identified were not stated clearly in the paper. Did the authors use H3K27ac or Mediator ChIP-seq data for the super-enhancer identification analyses?

RESPONSE:

We apologize for not clarifying the details of identifying the super-enhancers. As predicted by the reviewer, we had used H3K27ac ChIP-seq data.

Briefly, we used human super enhancer database-SEdb (PMID:30371817, <http://www.licpathway.net/sedb/index.php>) which curates and processes H3K27ac ChIP-seq data sets from publicly available resources and further computes super-enhancers using the ROSE pipeline. Specifically, we used the super-enhancer information identified from a MCF7 H3K27ac ChIP-seq data set from GSE57436. The recognized super enhancer at *KRT14/16/17* region is ranked #25 among all 210 super enhancers.

EDITS:

Page 34, line 709

We added a subsection “Super-enhancer identification” in the methods section.

“Super-enhancer identification:

Super-enhancers were identified from a human super enhancer database-SEdb (<http://www.licpathway.net/sedb/index.php>) which curates and processes H3K27ac ChIP-seq data sets from publicly available resources and further computes super enhancers using the ROSE pipeline. Specifically, we used the super enhancer information called from a MCF7 H3K27ac ChIP-seq data set from GSE57436. The recognized super enhancer at *KRT14/16/17* region was ranked #25 among all 210 super enhancers.”

6. Considering the *ESR1* mutant is associated with the BCK activation, immune activation, and better survival rate in this manuscript, it seems contradictory to the current knowledge that ER mutant is a bad predictor for patient survival. It will be appreciated if the authors can discuss this more in the Discussion section.

RESPONSE:

We thank the reviewer for raising this point. As the reviewer acknowledged, *ESR1* mutations are widely reported to be associated with poor prognosis, due to its known effect on ligand-independent activation of ER and gain of function of metastatic features. We agree with the reviewer that the concurrent induction of BCK expression and immune activation in *ESR1* mutant tumors causes a complex scenario, with basal-ness being associated with poor outcome and immune infiltration being associated with improved outcome. However, there is also data showing that increased immune infiltration is associated with worse outcome in ER+ disease (PMID: 31391067, 28859291), and indeed our analysis in ER+ primary tumors, both increased BCK expression and immune activation are linked to favorable outcomes. Clearly, additional studies are required to solidify and understand this complex data.

We have edited the Discussion accordingly and have also stressed that a major point of our findings is the discovery of a novel vulnerability of *ESR1* mutant breast cancer. It will be important to test the hypothesis *ESR1* mutant breast cancers are sensitive to immune therapy.

EDITS:

Page 23, line 454

We added the following sentence into the discussion section regarding to this point.

“Nevertheless, our data suggest the enhanced immune activation in *ESR1* mutant breast cancers as a novel vulnerability. There is data showing enhanced immune filtrations were associated with worse outcome of ER+ breast cancer⁸⁶, opening up the possibility that BCK-associated immune alterations might contribute to the inferior outcome of patients with *ESR1* mutant breast cancer. The undoubtedly complex role of immune infiltrates in ER+ breast cancer, particularly in the setting of *ESR1* mutant disease, requires further thorough investigation.”

Other Edits:

1. We added the following authors due to their significant contributions to the revision of this manuscript.

Olivia McGinn: Helped with clinical samples IF and IHC validation.

Sayali Onkar/ Caleb Lampenfeld/ Tullia C. Bruno/ Dario A.A. Vignali: Helped with experiments to identify role of S100A8/A9 in ER+ breast cancer.

2. Page 35, line 743

We added names of the following people into the Acknowledgement section due to their technical support during the work for the revision.

Jagmohan Hooda, Christy Smolak: Helped with animal study

Peter Lucas: Helped with pathological characterization of mouse tumor section

Alana Welm: Provide HCI-013EI PDX model for additional validation

3. Page 34, line 717

We added a section named “Statistical Analysis” in the method section as per the journal’ s request on the Reporting Summary.

“Statistical Analyses

All statistical analyses were specified at the corresponding figure legends. Two-side test was applied to all the analysis.”

4. Page 35, line 737

We deposited scripts associated with this study into Code Ocean and added a “Code Availability” section as per the decision letter.

“Code Availability

R script associated with this study was deposited into Code Ocean with DOI 10.24433/CO.0627595.v1 and will be publicly available upon publication”

5. Page 29, line 588

We moved the description of Immunofluorescence and Immunohistochemistry from the Supplementary Materials and Methods to the Main Materials and Method since the relevance of data generated by these two technologies increased in the revision manuscript.

Immunofluorescent Staining

MCF7 cells were hormone deprived and seeded on coverslips. After desired treatments, cells were fixed with 4% paraformaldehyde and blocked with 3% BSA solution plus 0.1% tritonX-100. Primary antibody against CK8 (Abcam, ab53280), CK5 (Abcam, ab52635), CK16 (Abcam, ab76416) and CK17 (Cell Signaling Technology, #4543) was applied to stain the cells. For counterstaining, CK16 (Santa Cruz, #53255) and ER (Licor, 6F11) mouse monoclonal antibodies were used to combine with above-mentioned rabbit CK5/16/17 antibodies. Secondary Alexa Fluor 488 or 546-conjugated antibodies (Thermo Scientific, A16079 & A11018) and Hoechst (Thermo Scientific, #62249) were used following primary antibody incubation. Coverslips were mounted and images were taken using

fluorescence microscope (Olympus, CZX16) under objective of 20X. CK5/16/17 positivity quantification was performed by dividing cells with full cytoskeleton CK expression to total cell numbers of each image. For ER counterstaining quantification, ER signal intensity was quantified using ImageJ for each CK positive cell and five proximal CK negative cells.

For staining on tissues, samples were fixed in 10% buffered formalin. Tissue was processed, paraffin embedded, and cut into 5- μ m sections. After high-temperature antigen retrieval in citrate buffer (pH 6.0). Sections were permeabilized with 0.1% Triton-X100 in PBS and blocked with 10% normal goat serum for 1h. Sections were stained with primary antibodies specific for CK5 (rabbit, 1:200, ab75869, Abcam), CK17 (rabbit, 1:100, ab109725, Abcam), PR (M3569, mouse, 1:100, Agilent) EpCAM (rabbit, 1:100, ab71916, Abcam), CD45 (rabbit, 1:200, ab10558, Abcam), or S100A8/9 (mouse, 1:200, NBP1-60157, Novus Biologicals, Littleton, CO, USA) for 2h at RT. Sections were stained with secondary antibodies Alexa Fluor goat-anti-mouse 488 (1:200, A-11001, Life Technologies, Rockville, MD, USA) and Alexa Fluor goat-anti-rabbit 647 (1:200, A-21245, Life Technologies) for 1h at RT. Sections were counterstained with Hoechst dye (1:2000 in PBS, Life Technologies). Slides were mounted using Fluoro-Gel Mounting Medium with Tris buffer (1798510, Electron Microscopy Sciences, Hatfield, PA, USA). Slides were imaged using an Olympus IX83 fluorescent microscope or a Nikon A1R confocal microscope.

Immunohistochemistry Staining

Tissue sections were processed as above and were stained with antibodies specific for CK5 (rabbit, 1:200, ab75869, Abcam, Cambridge, UK) and CK17 (rabbit, 1:100, ab109725, Abcam) for 2h at RT. Sections were blocked using HRP Blocking Reagent (Abcam) EnVision+/HRP Visualization (Agilent, Santa Clara, CA, USA) and DAB substrate kit (Agilent) were used to visualize staining. Sections were counterstained with hematoxylin and mounted with Permount Mounting Medium (Fisher Scientific, Waltham, MA, USA). Representative photographs were taken under a light microscope at 20X magnification.

Reviewers' Comments:

Reviewer #1:

Remarks to the Author:

The authors have provided a well-constricted and comprehensive response to the queries posed which have improved the manuscript. I have no further comments and recommend acceptance of the study.

Reviewer #2:

Remarks to the Author:

Authors have added a lot more data to support their main conclusion.

For comment 1: I agree that this manuscript "there are several novelties we would like to highlight based on our results."

For comment 2: I am very satisfied for additional data generated for GR.

For comment 3: Authors have performed a nice cytokine assay to address the question on MDSCs. S100A8 or S100A9 modulation by inhibitors did not significantly modulate cytokines expression suggested that these two proteins could serve biomarkers but not useful target for clinical application. ER is a strong transcriptional factor to modulate many cytokine expression. I would suggest that authors mention these negative data in the discussion. These data suggest that S100A8 and S100A9 are two potent biomarkers but may need other proteins to cooperate together to affect microenvironment including MDSCs.

I have suggested to perform a co-culture experiment by mixing ESR1 Mutant or WT cell lines with human PBMCs. The goal is to compare mutant to WT cells. Mutant cell should activate T cell proliferation in PBMC population based on your data. You could reverse the phenotype by knockdown S100A8 or S100A9. This experiment could be considered for next paper.

For comment 4. I really appreciate the authors' effort by performing in vivo experiment to address the role of S100A8 and S100A9 in immune activation phenotype. Unfortunately authors could not demonstrate the statistical significance of the two groups. However, the downregulation of CD45 could add value to the main conclusion. S100A8 and S100A9 could serve as the key players for immune activation in ESR1 mutant models.

Overall, I am satisfied for authors additional data.

More work still need to be done to better understand the role of S100A8 and S100A9 in ESR1 metastatic model and how ESR1 mutant genotype or phenotype could affect microenvironment in order to promote metastasis.

Reviewer #3:

Remarks to the Author:

The authors have addressed all the concerns that I have raised during my first review and this manuscript has been greatly improved. Good job!

Point-to-point reply to reviewer comments for Nature Communications manuscript (NCOMMS-20-51419B)

We greatly appreciate the in-depth and constructive comments from all three reviewers on our revised manuscript. Below is a point-to-point reply towards each single point. Reviewers' comments are in italic, our **direct responses are in red** and our **corresponding edits to the manuscript files are in blue** for readability.

Reviewer #1 (Remarks to the Author):

The authors have provided a well-constricted and comprehensive response to the queries posed which have improved the manuscript. I have no further comments and recommend acceptance of the study.

RESPONSE:

We thank the reviewer for the in-depth comments on our original submission and we believe that our study has strongly improved upon addressing the reviewer's concerns and suggestions.

Reviewer #2 (Remarks to the Author):

Authors have added a lot more data to support their main conclusion.

For comment 1: I agree that this manuscript "there are several novelties we would like to highlight based on our results."

For comment 2: I am very satisfied for additional data generated for GR.

For comment 3: Authors have performed a nice cytokine assay to address the question on MDSCs. S100A8 or S100A9 modulation by inhibitors did not significantly modulate cytokines expression suggested that these two proteins could serve biomarkers but not useful target for clinical application. ER is a strong transcriptional factor to modulate many cytokine expression. I would suggest that authors mention these negative data in the discussion. These data suggest that S100A8 and S100A9 are two potent biomarkers but may need other proteins to cooperate together to affect microenvironment including MDSCs.

I have suggested to perform a co-culture experiment by mixing ESR1 Mutant or WT cell lines with human PBMCs. The goal is to compare mutant to WT cells. Mutant cell should activate T cell proliferation in PBMC population based on your data. You could reverse the phenotype by knockdown S100A8 or S100A9. This experiment could be considered for next paper.

For comment 4. I really appreciate the authors' effort by performing in vivo experiment to address the role of S100A8 and S100A9 in immune activation phenotype. Unfortunately authors could not demonstrate the statistical significance of the two groups. However, the downregulation of CD45 could add value to the main conclusion. S100A8 and S100A9 could serve as the key players for immune activation in ESR1 mutant models.

Overall, I am satisfied for authors additional data.

More work still need to be done to better understand the role of S100A8 and S100A9 in ESR1 metastatic model and how ESR1 mutant genotype or phenotype could affect microenvironment in order to promote metastasis.

RESPONSE:

We thank the reviewer to acknowledge our efforts in addressing these concerns and we totally agree that further investigations on the role of the S100A8/A9 pathway in the *ESR1* mutant tumor microenvironment is warranted in future studies. As the reviewer suggested in comment 3, we have

Supplementary Fig. 12I. Supernatants from human-derived monocytes were treated with media alone, 100ng/ml LPS, or 10µg/ml S100A8/A9 for 24 hours and analyzed by the Proteome Profiler Human Cytokine Array Kit. Images of array membranes are shown. Negative and positive control spots are labeled and increased cytokines are highlighted. This experiment was done once.

now added the negative data of S100A8/A9 recombinant protein treatment on human-derived macrophages as new Supplementary Figure 12I and mentioned it in the Results and Discussion sections. Briefly, we proposed that potential interaction with additional components from the tumor microenvironment is necessary for the activity of S100A8/A9 and hence highlighted the necessity of using *in vivo* instead of *in vitro* models in future studies.

EDITS:

1. Page 18, line 402

We added description of this negative data and modified the transition to the single-cell RNA-seq data analysis.

“However, *in vitro* stimulation of human-derived macrophages with S100A8/S100A9 purified proteins failed to induce cytokine production (Supplementary Fig. 12I), possibly due to required interaction of the S100A8/S100A9 heterodimer with additional factors from the tumor microenvironment.

To further elucidate the specific cell-cell communication by S100A8/S100A9 signaling in the tumor ecosystem, we analyzed RAGE and TLR4 signaling via measuring ligand and receptor expression in different cell types using single-cell RNA-seq data from two breast cancer metastases.”

2. Page 21, line 464

We added the following sentence into the Discussion section:

“Notably, we failed to detect effects of S100A8/S100A9 using an *in vitro* system, suggesting the need for more complex model systems including *in vivo* models. This will also allow the analysis of MDSC which have been described to play an important role in S100A8/A9 function^{80,83}”

3. Supplementary Fig. 12

We added this data as new Supplementary Figure 12I.

4. Materials and Methods

We added the method description of this cytokine array.

Cytokine array with human-derived macrophages

Human-derived monocytes were obtained from a leukopak from a healthy donor. Monocytes were treated with M-CSF (Peprotech, 300-025) for 5 days to differentiate into M0 macrophages. Cells were then treated with medium alone, 100 ng/ml lipopolysaccharide (LPS, Millipore Sigma, L4391) or 10µg/ml recombinant human S100A8/S100A9 heterodimer protein (R&D Systems, 8226-S8) for 24 hours. 700 µL of cell supernatant was harvested for each sample and centrifuged to remove particles. Supernatants were analyzed with the Proteome Profiler Human Cytokine Array Kit (R&D Systems, ARY005B) following manufacturer protocol. Briefly, membranes were blocked with blocking buffer

supplied by the kit. Samples were diluted with assay buffer as described in the manufacturers protocol and incubated overnight with membranes and antibody cocktail. Membranes were washed, incubated with Streptavidin-HRP buffer supplied in the kit and incubated for 30 minutes at room temperature. Arrays were washed and imaged by chemiluminescence using a BioRad ChemiDoc XRS+ molecular imager.

Reviewer #3 (Remarks to the Author):

The authors have addressed all the concerns that I have raised during my first review and this manuscript has been greatly improved. Good job!

RESPONSE:

We greatly appreciate the reviewer's acknowledgment of the improvement of this work, as well as the constructive comments brought up in response to our original submission.